# Preliminary Comparisons of Tender Shoots and Young Leaves of 12 Mulberry Varieties as Vegetables and Constituents Relevant for Their Potential Use as Functional Food for Blood Sugar Control

**DOI:** 10.3390/plants12213748

**Published:** 2023-11-02

**Authors:** Jia Wei, Yan Zhu, Tianbao Lin, Han Tao, Lei Chen, Zilong Xu, Zhiqiang Lv, Peigang Liu

**Affiliations:** 1Institute of Sericulture and Tea, Zhejiang Academy of Agricultural Sciences, Hangzhou 310021, China; weijia@mail.zaas.ac.cn (J.W.); zhuy@mail.zaas.ac.cn (Y.Z.); lintb@mail.zaas.ac.cn (T.L.); chenlei_0117@163.com (L.C.); xuzilong@zaas.ac.cn (Z.X.); lvzq@mail.zaas.ac.cn (Z.L.); 2Zhejiang Provincial Key Laboratory of Resources Protection and Innovation of Traditional Chinese Medicine, Zhejiang A&F University, Hangzhou 311300, China; taohan@zju.eu.cn

**Keywords:** nutritional composition, mulberry varieties, blood sugar control

## Abstract

Vegetables are essential for maintaining health and preventing diseases due to their nutrients and functional components. However, vegetables specifically designed for blood sugar control are limited. The mulberry tree (Morus) offers potential as a source of functional vegetables with blood-sugar-lowering properties, mainly attributed to 1-Deoxynojirimycin (DNJ). This study compared the nutritional composition and DNJ content in various edible parts of twelve mulberry tree varieties. Sensory evaluations were also conducted to assess sensory attributes. Interestingly, DNJ was found to show a positive correlation with sensory evaluations. Furthermore, the sugar content, particularly sucrose, was significantly higher in tender shoots than leaves, indicating tender shoots as a preferable choice for development as a functional food for blood sugar control. Finally, *VM 19* and *VM 22* are considered as good candidates for the mulberry vegetable using varieties after sensory evaluation and combining with the DNJ content. These findings provide valuable insights for future research into vegetable selections for blood sugar management and support the potential commercialization of mulberry leaf vegetables as functional food options.

## 1. Introduction

Types of vegetables play a vital role in a healthy diet due to their numerous health benefits. Vegetables are rich in essential nutrients, including vitamins, minerals, and dietary fiber, which are necessary for maintaining optimal health and preventing various diseases for the consumer [1,2]. Beyond their traditional nutritional value, functional components in vegetables have gained increasing attention due to their unique properties that contribute to various physiological functions in the body. These compounds exhibit a range of activities, including antioxidant, anti-inflammatory, anticancer, antimicrobial, and immune-enhancing properties [3,4]. Antioxidants such as vitamin C (VC), and vitamin E (VE), are vital functional components found in vegetables that help protect the body against oxidative stress caused by free radicals. Anti-inflammatory compounds present in vegetables, such as flavonoids, help modulate the body’s inflammatory response [5,6]. Certain vegetables contain phytochemicals that have demonstrated potential anticancer properties, such as sulforaphane in cruciferous vegetables and lycopene in tomatoes [7,8]. By gaining knowledge about the levels and functions of active components in different vegetables, individuals can make informed choices of vegetables that align with their specific health needs and conditions.

The global prevalence of individuals with high blood sugar levels, such as diabetes, is alarmingly high. This significant population faces a growing need for vegetables that can help regulate and reduce blood glucose levels. However, it is unfortunate that currently there is a limited availability of vegetables with specific blood-sugar-lowering properties. The development and discovery of more vegetables with such functionalities are crucial to meet the demands of the large number of people worldwide affected by high blood sugar. Incorporating these functional vegetables into their diets can provide individuals with effective tools for managing their condition and improving their overall health.

A mulberry tree (Morus) is a fast-growing and versatile economic tree species. The tender shoots and leaves of the mulberry tree are rich in nutrients, including proteins, fiber, vitamins, and minerals [9,10]. In addition to its nutritional components, like other vegetables, mulberry leaves are also rich in bioactive compounds such as flavonoids, polyphenols, and, most notably, its unique active ingredient, DNJ (1-Deoxynojirimycin). DNJ has been extensively proven to have blood-glucose-lowering properties [11,12]. Furthermore, numerous studies have demonstrated that mulberry leaves function as a functional food with anti-diabetic activity, as incorporating mulberry leaf products into one’s diet or consuming mulberry leaves can effectively reduce blood sugar levels. 

Mulberry leaves have a rich application history in Asian countries like China, Japan, and Korea, where they have been used both as food and for medicinal purposes. In fact, the Chinese Ministry of Health recognized their dual role in 2002 when they issued a ‘List of Items that are Both Food and Medicine’ [13]. This list includes items that are not only safe for consumption but also offer therapeutic and disease-preventive benefits, with mulberry leaves being one of the items featured. 

The young leaves of mulberry can be prepared in various ways as vegetables, such as soups, stir-fried dishes, and salads [14,15]. The interest in incorporating mulberry leaves as a vegetable into the diet is increasing, providing more options for diversified and healthy eating. However, the considerable variations in nutrient content, DNJ levels, and sensory evaluation among different mulberry tree varieties and parts underscore the importance of conducting further research. This will enable us to fully explore their potential and make informed choices when incorporating mulberry leaves as a valuable vegetable option.

The present study aims to investigate and compare the nutritional composition and DNJ content of twelve differential mulberry tree varieties, focusing on various edible parts, including tender shoots and leaves. Additionally, we conducted sensory evaluations to assess the organoleptic qualities of these mulberry leaf vegetables. This comprehensive analysis allows us to identify varieties with optimal sensory attributes and nutritional profiles. The findings hold significant implications for the potential large-scale commercialization of mulberry leaf vegetables.

## 2. Materials and Methods

### 2.1. Mulberry Sample Preparation

The samples of tender shoots (top three leaves) and young leaves (leaves at the five or six position) of mulberry trees (10 years old) were collected from twelve mulberry varieties planted in the mulberry fields of the Institute of Sericulture and Tea, Zhejiang Academy of Agricultural Sciences, on 25 May 2021. All the mulberry trees of differential varieties were planted in March 2010 and cultivated in similar agroclimatic conditions, with an altitude of 20 m above mean sea level (latitude 30.314 N and longitude 120.196 E). 

Tender shoots and leaves of differential mulberry varieties were sampled from thirty individual trees of each variety according to the same sample standards on a cloudless day. After the sampling, most parts of the tender shoot and leaf samples were used for sensory evaluation, and the rest of the tender shoot and leaf samples were frozen in liquid nitrogen immediately and stored at −80 °C until a nutritional and antioxidant activity analysis. Before the nutritional and antioxidant activity analysis, parts of pre-frozen tender shoot and leaf samples were freeze-dried at −100 °C for 36 h using a Labconco FreeZone freeze-dryer (Labconco, Kansas City, MO, USA). After the freeze-dried treatments, the freeze-dried samples were then ground to powder, packed, and stored at −80 °C until use.

### 2.2. Determination of the Content of Soluble Protein (SP), Sucrose, Glucose, and Fructose 

The SP content was determined according to the Bradford method as previously described [16] by using bovine serum albumin. A 0.2 mL SP extract was added to 2.4 mL of a Coomassie Brilliant blue G 250 solution and 9.6 mL of deionized water. After incubation for 10 min at room temperature, the mixture was measured with spectrophotometry at 595 nm. The SP contents were expressed as mg/100 g of dry weight (DW). 

The soluble sugar determination of tender shoots and leaves of differential mulberry varieties was carried out with HPLC according to the method described earlier by Zhao et al. [17] with slight modifications. In total, 0.1 g of freeze-dried sample powder was homogenized in 2.5 mL of 80% ethanol, and then incubated in a water bath at 65 °C for 20 min. After centrifugation, the supernatant was collected, and the precipitate was extracted at another time. Then, supernatants were combined and concentrated, and the contents of soluble sugars were determined using an HPLC system consisting of a Waters 600 separations module connected to a Waters 2414 RI detector (Waters Corp., Milford, MA, USA). The sugars were separated on a Sugar-PaK I column (300 mm × 6.5 mm, 5 μm; Waters, New York, NY, USA) with a mobile phase (deionized, bacteria-free water containing 0.0001 M calcium EDTA) at a flow rate of 0.5 mL/min. The injected sample quantity was 20. These three types of soluble sugar contents were quantified according to the external standard method. Each sample was confirmed with three biological repeats. The sugar concentration was expressed as mg/g of DW. 

### 2.3. Determination of the Content of Anthocyanidin, Total Flavonoids (TF), and Total Polyphenols (TP)

The anthocyanidin content was tested by using the pH differential method. Approximately 0.5 g of freeze-dried sample powder was added to 10 mL of methanol containing 1% HCl (*v*/*v*) and sonicated in the dark for 1 h at room temperature. After centrifuging, the absorbance of the supernatants was measured for absorbance at 530 and 657 nm, respectively. The anthocyanin content was estimated using the following formulas and expressed as mg of cyanidin-3-glucoside (CG) equivalent per g of extract (mg of CG/g of DW): The anthocyanin content = [A530 − A657/4] × V × M/(ε × m).
where V is the extract volume (mL), ε is the molar extinction coefficient of cyanidin-3-glucoside at 530 nm (29,600), M is the molecular weight of CG (449.38 g/mol), and m is the mass of the sample powder for extract use. The TF content was determined by using the sodium-nitrite–aluminium-nitrate colorimetric method. Approximately 0.1 g of the freeze-dried sample was extracted with 1 mL of 80% methanol with ultrasonic treatment for 30 min. After centrifugation, 0.5 mL of the extract was diluted with 4.5 mL of 80% ethanol. Then, 6 mL of the diluted supernatant was mixed with 1 mL of 5% NaNO_2_ and incubated for 6 min. And then 1 mL of 10% Al(NO_3_)_3_ was added and incubated for 6 min. Then, 10 mL of 1 M NaOH was added and the final volume was made up to 25 mL with distilled H_2_O, and the mixture incubated at room temperature for 15 min. The absorbance was read at 500 nm. TF content was expressed as milligrams (mg) of quercetin equivalent (QE) per gram (g) of extract (mg of QE/g of DW).

The TP content was measured in accordance with the Folin–Ciocalteau method. The sample powder (0.2 g) was homogenized with 2 mL of 75% ethanol with ultrasonic treatment for 30 min and subsequently centrifuged at 8000× *g* for 2 min. A 0.3 mL diluted supernatant was mixed with 4 mL of a 10% Folin–Ciocalten reagent. After mixture incubation for 2 min in the dark, 2 mL of a 7.5% Na_2_CO_3_ solution was added and incubated for 1 h at room temperature. After that, the absorbance of the mixture was measured at 765 nm and the TP content was expressed as mg of gallic acid (GAE) equivalent per g of extract (mg of GAE/g of DW).

### 2.4. Determination of DNJ Content and VC Content

DNJ content was determined by using an HPLC method previously reported by Kim et al. [18] with some modifications. In total, 2 g of sample powder was dissolved with 50 mL of 65% ethanol, followed by vortexing for 1 min, and then treated ultrasonically for 40 min. After centrifugation at 8000× *g* for 30 min, 10 μL of the derivatized tender shoot and leaf samples of mulberry was injected into an HPLC system (Waters, New York, NY, USA). DNJ content was measured with a SunFireTM C-18 (250 mm × 4.6 mm) 5μm column with a Waters 2475 photodiode array detector. The determination was carried out at the column temperature of 25 °C with the detection wavelength of 254 nm. The sample was eluted with a mobile phase of 0.05% aqueous acetic-acid–acetonitrile (65:35, *v*/*v*) at a flow rate of 1.0 mL/min for 30 min. The DNJ concentration in samples was calculated by using the equation of the calibration curve. All assays were run in triplicate. 

VC content was detected with the modified high-performance liquid chromatography (HPLC) method suggested by Cemeroglu [19] with some modifications. VC in fresh samples (1 g) was extracted using 2% oxalic acid (10 mL), and then separated through a SunFireTM C-18 (4.6 mm × 250 mm, 5 μm) chromatographic column with the mobile phase of 0.2 mol/L of ammonium-acetate-solution–acetonitrile (40:60), and carried out at the flow rate of 1 mL/min. The column temperature was 30 °C, and the detection wavelength was 280 nm. The VC concentration in samples was calculated on the basis of the calibration curves of VC. All assays were run in triplicate.

### 2.5. Determination of Total Antioxidant Activities

The total antioxidant activities of tender shoot and leaf samples of differential mulberry varieties were measured using three methods, namely the Diphenyl-1-picrylhydrazyl (DPPH) method, 2,2-Azino-bis-3-ethylbenzothiazoline-6-sulfonic Acid (ABTS) method, and ferric reducing antioxidant power (FRAP) method. 

DPPH assay: This procedure was conducted according to the method described earlier by Brand-Williams et al. [20] with slight modifications. In total, 1.0 mL of 80% methanol was added to freeze-dried sample powder (0.1 g), with ultrasound for 5 min, and then centrifuged at 8000× *g* at 4 °C for 5 min. A 0.4 mL supernatant in 0.6 mL of 0.1 mM DPPH in methanol was vortexed sufficiently and stored in the dark for 30 min at ambient temperature. And then the mixture was centrifuged at 8000× *g* for 5 min at 4 °C. The absorbance values were determined using a spectrophotometer (Infinite M200, Tecan, Seestrasse, Switzerland) at 517 nm. The group without mulberry leaf samples and containing all of the chemical reagents described above was used as the control samples. The radical scavenging activities of the DPPH radical were calculated with the following formula:DPPH radical scavenging activities = (A_1_ − A_0_)/A_0_ × 100
where A_0_ is the absorbance of control samples, and A_1_ is the absorbance of mulberry leaf samples.

FRAP assay: The FRAP assay procedure was carried out according to the modified method of Benzie and Strain [21]. Approximately 1 g of freeze-dried sample powder was mixed with 4 mL of a PBS solution and incubated for 5 min on ice. After centrifugation at 8000× *g* for 5 min at 4 °C, 5 µL of the extract supernatant was mixed with 180 μL of the FRAP working solution and heated at 37 °C for 5 min in a water bath. The absorbance was then measured by using an Infinite M200 spectrophotometer at a wavelength of 593 nm. The final result was expressed as the concentration of antioxidants having a ferric reducing (FRR) ability in 1 g of the sample (l M/g).

ABTS assay: A total antioxidant activities assay kit with the ABTS method was used to determine the ABTS radical scavenging activity [22]. The freeze-dried sample powder (0.5 g) was mixed with 4.5 mL of 0.9% sterile saline and grinded for 5 min on ice, then centrifuged at 8000× *g* for 5 min at 4 °C. The ABTS working solution was made from the ABTS solution mixed with the oxidant solution (1:1, *v*/*v*), which was incubated in the dark at room temperature for 12–16 h. The working solution was diluted using 80% (*v*/*v*) ethanol. An aliquot of the supernatant (100 µL) was mixed with 3.9 mL of the diluted ABTS radical cation solution. After a reaction at room temperature for 6 min, the absorbance of the mixture was determined at 734 nm with an Infinite M200 spectrophotometer. The ABTS radical scavenging activities of the sample were expressed as the Trolox-E (TE) activity capacity. 

### 2.6. Quantitative Descriptive Analysis (QDA) for Mulberry Vegetable

The QDA was conducted in a sensory laboratory to evaluate sensory characteristics of the tender shoot and leaf samples used for the mulberry vegetable. The tender shoot and leaf samples of differential mulberry varieties were tested by a sensory evaluation group of 12 food sensory evaluators (7 females and 5 males, aged between 27 and 63 years, and defined based on the following standards of ISO 8586: 2012) [23] who have good experience in vegetable sensory evaluation. The sensory evaluation was performed at the standard sensory laboratory of the Institute of Sericulture and Tea, Zhejiang Academy of Agricultural Sciences, China. The standard sensory laboratory was prepared for work following guidelines given in ISO 8589: 2007 [24]. The 9-point hedonic scale (1: Dislike very much and 9: Like very much) [25] combined with a previously reported sensory quality evaluation standard on mulberry leaf [26] was used to evaluate sensory evaluation for the tender shoots and leaves of mulberry. 

In the sensory evaluations, color, texture, flavor, and taste of all tender shoot and leaf samples of differential mulberry varieties were assessed according to the following standard: color—normal color, slightly tawny (1–3), light and bright in color (4–6), bright in color (7–9); texture—rough (1–3), soft (4–6), crispy (7–9); flavor—slight mulberry leaf flavor (1–3), stronger mulberry leaf flavor (4–6), rich mulberry leaf flavor (7–9); taste—bitter (1–3), slightly bitter (4–7), slightly sweet (7–9). The average scores of sensory attributes based on the scores given by twelve sensory evaluators were provided as evaluation results.

### 2.7. Statistical Analysis and Pearson Correlation Coefficients (PCC)

All data are expressed as the mean ± standard deviation (SD), and the data were analyzed using SPSS version 23.00 (SPSS Inc., Chicago, IL, USA), and statistical significance was evaluated by using an analysis of variance (ANOVA) and least significant difference (LSD) test and set at *p* < 0.05, and the coefficient of variation (CV) of the index between varieties was calculated with the formula CV = mean/SD. Evaluations of total antioxidant activities and sensory factors were performed by using the subordinate function value method (SFVM) and normalized with the formula R (x_i_) = (x_i_ − x_min_)/(x_max_ − x_min_), where x_max_ and x_min_ are the maximum and minimum of x_i_, and the comprehensive evaluation score of total antioxidant activities or sensory factors is the mean score (R_x_) of their determination data or sensory evaluation indexes. The Pearson Correlation Coefficients (PCC) analysis was performed using origin 2021 software.

## 3. Results

### 3.1. The Shape, Color, and Carbohydrates Profile in Tender Shoots and Leaves of Differential Mulberry Varieties 

The shape and color in tender shoots and leaves of differential mulberry varieties are shown in Figure 1. As shown in Figure 1, the tender shoots’ color of most mulberry varieties is light green, and a small part of them are mauve pale (*VM 1*) and yellow green (*VM 23*). The leaves’ color of varieties can be expressed as light green (such as *VM 13*), green (such as *VM 1*), and dark green (such as *VM 10*), and the surfaces of leaves vary enormously between differential varieties, from relatively smooth surfaces of amorphous wax (such as *VM13*) to highly rough surfaces where the surface of the leaf is (such as *VM 10*). 

From Figure 2, it clearly shows that the SP content in tender shoots is higher than that in leaves of every mulberry variety; mean SP content of tender shoots is more than four times that of leaves, especially in *VM 5*; SP content of tender shoots is more than twenty times that of leaves. And we also found that the varieties’ differences of soluble protein content in leaves (CV = 20.37) are bigger than those of tender shoots (CV = 6.07) (Figure 2). The content of sucrose, glucose, and fructose in tender shoots and leaves varies among differential mulberry varieties (Figure 2). Evidently, the mean content of sucrose and glucose in leaves was about eight times and four times higher in tender shoots, respectively; on the contrary, the mean content of fructose in leaves was slightly lower than that in tender shoots. There are stark differences in the content of sucrose, glucose, and fructose in tender shoots and leaves of differential mulberry varieties, and the coefficients of variation (CV) of six indices all exceed 20% (Appendix A). Additionally, we found that the sucrose content in leaves of differential mulberry varieties is all higher than that in tender shoots.

### 3.2. Polyphenolic Compound Content in Tender Shoots and Leaves of Differential Mulberry Varieties

The content of differential types of polyphenolic compounds, such as anthocyanidin, TF, and TP, was detected in the tender shoots and leaves of twelve mulberry varieties (Table 1).

According to the statistical data presented in Table 1, it can be seen that the variety differences of the content of the anthocyanidin, TF, and TP in tender shoots were higher than those in leaves. In this study, we observed that the CVs of the TF content in tender shoots or leaves were all over 10%, and the variety differences (CV = 15.93%) in the tender shoots were higher than those in the leaves (CV = 11.01%). The TF content was found to be highest in tender shoots of *VM12* and *VM13* and to reach 24.59 mg/g of DW, and lowest in leaves of *VM 22* (13.62 mg/g of DW). The mean content of anthocyanidin in tender shoots is higher than that in leaves; in tender shoots, the anthocyanidin content is rich in *VM18* and *VM 22*, and more than 3.4 mg/g of DW; and *VM1* and *VM10* are the two leaves of anthocyanidin-rich mulberry varieties. The TP contents in the tender shoot (from 44.00 to 49.38 mg/g of DW) were all higher than those in the leaves (from 39.92 to 46.80 mg/g of DW) of each mulberry variety, and there were little differences among varieties on TP content in tender shoots and leaves; the CVs for differential varieties were all lower than 6%.

### 3.3. DNJ and VC Content in Tender Shoots and Leaves of Differential Mulberry Varieties

The content of DNJ and VC in tender shoots and leaves of differential mulberry varieties was tested in this study (Figure 3). The DNJ content of tender shoots and leaves also greatly varied with mulberry varieties, especially in leaves with a CV of 55.92% (Figure 3a, Appendix A), and the DNJ content of the leaves of the highest-DNJ rich mulberry variety (*VM 13*) was nearly five times that of the mulberry variety with the lowest DNJ content (*VM 1*). Of differential mulberry varieties, DNJ content of tender shoots ranged from 4.92 to 8.92 mg/g of DW, and that of leaves varied greatly in the range of 2.08 to 9.84 mg/g of DW among varieties (Appendix A), and the DNJ content of most varieties in tender shoots (10/12) rose to 6 mg/g of DW; however, there are only four varieties with more than a 6 mg/g of DW DNJ content. Notably, our analysis identified *VM7*, *VM13*, and *VM19* as major DNJ accumulators, thereby showing that those varieties can be recommended for consumption or as breeding partners to increase DNJ levels in mulberry.

The VC content results (Figure 3b) indicated that mulberry leaves are rich in VC, and its content in tender shoots varied in the range of 129.97 to 153.68 mg/100 g of FW, and it ranged from 117.80 to 137.00 mg/100 g of FW in leaves. In addition, it clearly shows that the VC content in tender shoots was higher than that in leaves among most varieties (11/12) in the present study; only of *VM 18* is VC content in tender shoots lower than that in leaves. And there were subtle differences in differential varieties with the CVs all being lower than 6% (Appendix A), indicating that differences between varieties about VC content are weak. 

### 3.4. Total Antioxidant Activities of Tender Shoots and Leaves in Differential Mulberry Varieties

In the present study, the total antioxidant activities in tender shoots and leaves of differential mulberry varieties were determined with three complementary assays, namely FRAP, ABTS, and DPPH. The total antioxidant activity results for tender shoots and leaves in differential mulberry varieties are presented in Figure 4. From Figure 4a–c, we found that the mean values of the DPPH radical scavenging activities, FRAP values, and ABTS radical scavenging activities of tender shoots of twelve varieties were all higher than the mean values of those in leaves (Appendix A). The differences between varieties of DPPH radical scavenging activities (CV of tender shoots, leaves = 15.64, 20.53) were higher than those of ABTS radical scavenging activities (CV of tender shoots, leaves = 8.64, 10.39) and FRAP values (CV of shoots, leaves = 9.51, 9.61), indicating that ABTS radical scavenging activities varied more with mulberry varieties. And total antioxidant activity activities have similar trends in mulberry varieties; DPPH free radical scavenging activities, ABTS radical scavenging activities, and FRAP values were higher in *VM 7* and *VM 19*, while lower in *VM 1* and *VM 12*.

It can be seen that the comprehensive evaluation score of antioxidant activities is significantly positively correlated with ABTS radical scavenging activities, DPPH radical scavenging activities, and FRAP values of tender shoots (R = 0.78, 0.92, 0.91; *p* < 0.05) from the PCC analysis results shown in Figure 4. The results show that the FARP values, and DPPH radical scavenging activities, have a significant relation with the overall score of antioxidant activity of leaves, with the R score of 0.8 (*p* < 0.01) and 0.84 (*p* < 0.01), respectively, while there is a lower relation between the ABTS radical scavenging activities and the comprehensive evaluation score of total antioxidant activities (R = 0.25, *p* < 0.05) (Figure 5). From Figure 5, ‘DNJ content’ was found to have the highest positive relation with the comprehensive evaluation score compared to other compound indexes (R = 0.46, 0.64), exhibiting close association with the total antioxidant activities of tender shoots or leaves. Especially DNJ content in leaves show a significant positive relation with the comprehensive evaluation score of total antioxidant activities (*p* < 0.01). 

### 3.5. Sensory Quality Character Indexes of Tender Shoots and Leaves in Differential Mulberry Varieties

The sensory scores about the color, texture, flavor, taste, and comprehensive evaluation score of tender shoots and leaves in differential mulberry varieties are presented in Table 2. As shown in Table 2, some of the tender shoots’ color of mulberry is bright in color, some are light and bright in color, and two varieties (*VM 10*, *VM 16*) are a normal color that is slightly tawny; while leaves of a part of the varieties are bright in color, and most of them (8/12) are light and bright in color, only one variety is a normal color that is slightly tawny; therefore, CV of the color of tender shoots (36.40%) was higher than that of leaves (25.46%). The texture of tender shoots of most mulberry varieties (9/13) is soft, and the other three varieties are crispy and with a higher sensory score, while leaves of samples of most mulberry varieties (9/13) are rough and only three varieties are soft; these results showed that tender shoots are crispier and softer than leaves. Variety differences of the flavor score in tender shoots were significantly lower than in leaves, and CVs of them were 23.09% and 29.01%, respectively. It is clearly seen that there are five varieties that exhibit stronger mulberry leaf flavor of leaf samples and only one variety in tender shoots, and this indicates that the mulberry leaf flavor of leaves is stronger than tender shoots. Parts of tender shoot samples exhibit a slightly sweet taste and obtain a high sensory score, some of them are slightly bitter, and part of them (3/12) are bitter and obtain a lower sensory score; in leaves, most varieties (8/12) have a bitter taste and the sensory score is lower, and the other four varieties exhibit a slightly sweet taste and the sensory score is higher. The mean taste score of leaves is significantly high compared to that of tender shoots, and taste scores in tender shoots and leaves all have great variety differences, and the CVs separately reach to 38.20% and 72.62%. 

Of tender shoots, *VM7*, *VM19*, and *VM22* have been considered as the three varieties with a higher sensory evaluation value for vegetable use, and the comprehensive evaluation scores (sensory) were ranked in the top three of twelve varieties (Table 2), while the leaves of *VM 13*, *VM19*, and *VM22* obtained higher scores in sensory evaluation and the comprehensive evaluation scores (sensory) were ranked in the top three of twelve varieties (Table 2). Additionally, it can be seen that the tender shoots of *VM7*, *VM19*, and *VM22*, and the leaves of *VM 13*, *VM19*, and *VM22*, have a large content of DNJ, particularly the content of the leaves of *VM 13*, *VM19*, and *VM22* that was significantly higher than that in the leaves of other varieties (Figure 3a). Thus, *VM7*, *VM13*, *VM19*, and *VM22* could be chosen as the better-vegetable-use mulberry varieties.

According to the profile and content of carbohydrates, SP content, polyphenolic compounds, DNJ, and VC in tender shoots or leaves of mulberry, we further studied the relationship between sensory quality character indexes and the content of carbohydrates, SP, polyphenolic compounds, DNJ, and VC, and PCC results are shown in Figure 6. It is clearly shown that comprehensive evaluation sensory quality character indexes of tender shoots or leaves were all stronger related to texture and taste, and the correlation coefficients were all exceeding 0.9 (R of tender shoots = 0.96, 0.96 (*p* < 0.01); R of leaves = 0.91, 0.94 (*p* < 0.01)), indicating that texture and taste were two important quality characteristics and major factors affecting sensory perception and consumer acceptance of mulberry vegetables. Additionally, the correlation coefficients of the comprehensive evaluation score and color or flavor were all higher than 0.7 but lower than 0.8 (R tender shoots = 0.78, 0.73 (*p* < 0.01); R of leaves = 0.79, 0.78 (*p* < 0.01)), and were less than texture and taste.

Of the content of carbohydrates, SP, polyphenolic compounds, DNJ, and VC, in tender shoots, there are five indexes that displayed a positive correlation with the overall sensory score, and the correlation of SP content, VC content, and DNJ content is weak (R = 0.4, 0.28, and 0.35), and the correlation coefficients (R) of fructose content and DNJ content are lower than 0.1, while four indexes in leaves have a negative correlation with the overall sensory score. In leaves, there are six indexes that showed positive correlation with the comprehensive evaluation sensory score, and most of them have a weak positive correlation (R ranged from 0.08 to 0.21); only DNJ content has a strong correlation with a correlation coefficient (R) of 0.85, while three indexes in tender shoots have a negative correlation with the overall sensory score, and the correlation of VC content is moderate with a correlation coefficient (R) of −0.46.

## 4. Discussion

The present study aimed to investigate the differences in nutritional composition between tender shoots and leaves of mulberry tree varieties and their implications for differential target populations. The findings revealed significant variations in sugar content, composition, DNJ content, and antioxidant activity in the leaves among different mulberry tree varieties. However, no significant content differences were observed in other active substances such as polyphenols, flavonoids, anthocyanins, and VC. 

An excellent vegetable variety should have a high sensory evaluation score, with suitable color, texture, flavor, and taste, for obtaining a good sensory acceptability, while sensory acceptability is closely related to the nutritional composition of vegetables [26,27].

According to reports, secondary metabolites such as flavonoids, phenolic acids, terpenes, and glucosinolates are typically found in significantly higher levels in tender tissues of plants, such as shoot apical tissues, compared to matured leaves. This could be due to the intense growth of tender parts, which require a substantial amount of these active ingredients [18,21,28,29,30]. Our findings support this notion. We observed that tender shoots contained higher levels of active compounds, including polyphenols (such as flavonoids and anthocyanins), VC, and DNJ, compared to leaves. Consequently, the total antioxidant activities of tender shoots exceeded those of leaves. 

Soluble proteins and soluble sugars are two types of important carbohydrates and nutritional components of vegetables, and play important roles in the taste, palatability, and acceptability of vegetables [31,32]. And the sweet taste of a vegetable was indicated to be related to the concentration of sugars, and soluble proteins were found to elicit a sweet taste for humans [33,34]. When it comes to the carbohydrate profile, we observed higher levels of sucrose and glucose in leaves compared to tender shoots. Sucrose plays a crucial role as an energy and carbon source in plants, as it is synthesized through photosynthesis and subsequently transported and stored within the plant, and has an important effect on the taste and flavor of a vegetable [35,36]. And higher levels of SP in tender shoots compared to leaves were found in the present study. This significant disparity in sucrose content can be attributed to the lower photosynthetic ability of tender leaves in comparison to matured leaves. Similar findings have been reported for tea plants and headed cabbage [21,37]. 

As an important micronutrient in vegetables, VC plays important physiological and therapeutic roles [38]. In the present study, mulberry leaves were found to be rich in VC, and can be chosen as better VC resources; the VC content in tender shoots and leaves of mulberry was higher than many main vegetables, such as Brassica vegetables, okra, potatoes, green beans, spinach, and peas [39,40]. 

DNJ is a major bioactive compound of mulberry, and has been indicated to have a better effect on glycemic control in diabetes; also, DNJ was found to display a better antioxidant activity [11,41]. When we combined quality indicators with sensory evaluations, we were surprised to find that mulberry tree varieties with high DNJ content exhibited the strongest antioxidant activities and received higher sensory evaluation scores, suggesting that DNJ content can serve as an important reference indicator when selecting suitable mulberry tree varieties for consumption. Besides its known blood-sugar-lowering activity, DNJ has been shown to possess significant antioxidant properties [42,43]. However, currently, there are no reports linking DNJ content to taste perception. The substantial contribution of DNJ to sensory evaluations raises the need for further in-depth research to explore the underlying reasons. 

## 5. Conclusions

In conclusion, our findings indicate that the optimal selection of mulberry tree varieties varies based on specific needs. For researchers investigating treatments who need to control blood sugar levels, tender shoots of varieties with high DNJ content are recommended. These tender shoots have significantly lower sugar content and higher levels of DNJ compared to leaves, such as the tender shoots of *VM7*, *VM19*, and *VM22* in the present study, and this makes them suitable for researchers investigating to manage blood sugar levels effectively. For researchers investigating general health benefits or involved in the mulberry leaf vegetable industry, leaves with high DNJ content are an excellent choice. These leaves not only receive high sensory evaluation scores but may also offer various health benefits, such as the leaves of *VM 13*, *VM19*, and *VM22* in the present study. Furthermore, their larger biomass and higher yield make them economically advantageous and suitable for a wider range of applications among the general population or businesses. The abundance and higher production volume of leaves make them a more cost-effective option compared to tender shoots. Thus, *VM19* and *VM22* are recommended to be good candidates for vegetable-use varieties after the comprehensive analysis, in which tender shoots and leaves were all with higher sensory evaluation value and higher DNJ content.

## Figures and Tables

**Figure 1 plants-12-03748-f001:**
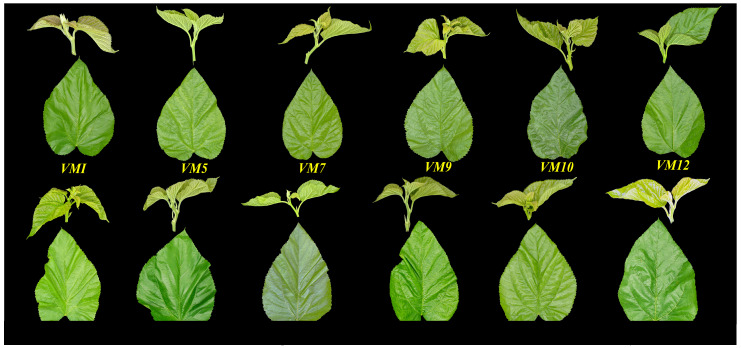
The shape and color of tender shoots and leaves of differential mulberry varieties.

**Figure 2 plants-12-03748-f002:**
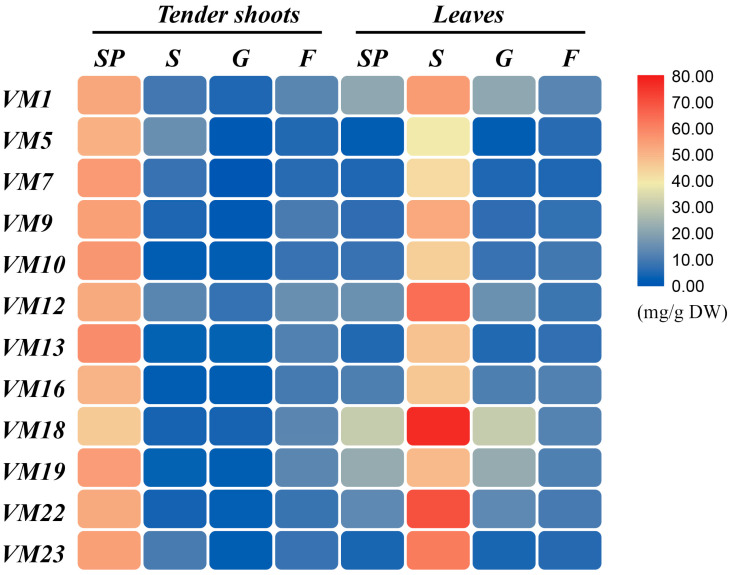
The carbohydrate profiles and protein of tender shoots and leaves of differential mulberry varieties. SP: soluble proteins; S: sucrose; G: glucose; F: fructose. The data are mean values of three replicates and standard deviation of the mean.

**Figure 3 plants-12-03748-f003:**
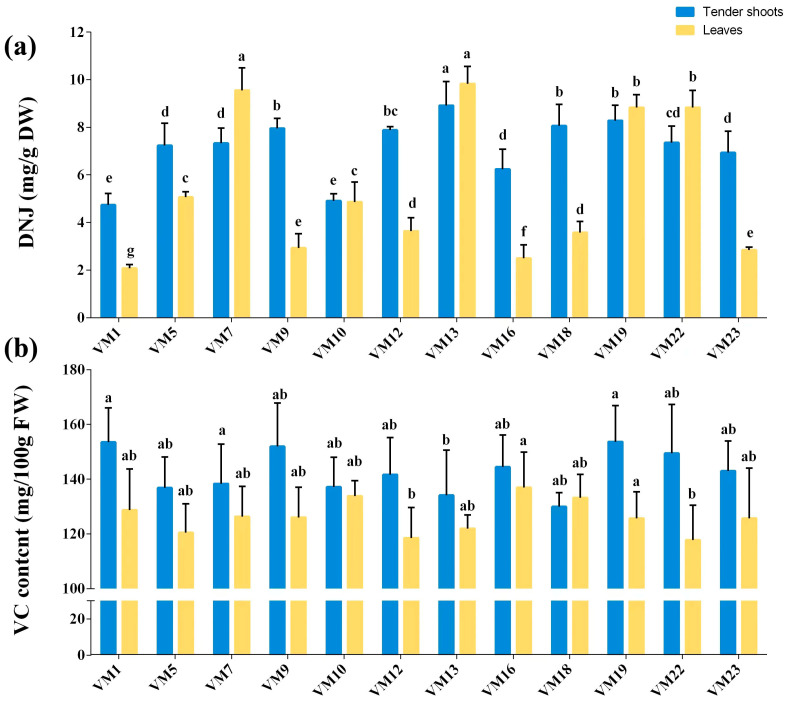
DNJ and VC contents in tender shoots and leaves of differential mulberry varieties. (**a**) DNJ content of tender shoots and leaves of differential mulberry varieties; (**b**) VC content of tender shoots and leaves of differential mulberry varieties. The data are mean values of three replicates and standard deviation of the mean, and the different small letter superscripts within the same group (tender shoot group or leaf group) of histograms represent significant differences at *p* < 0.05 (ANOVA and LSD test).

**Figure 4 plants-12-03748-f004:**
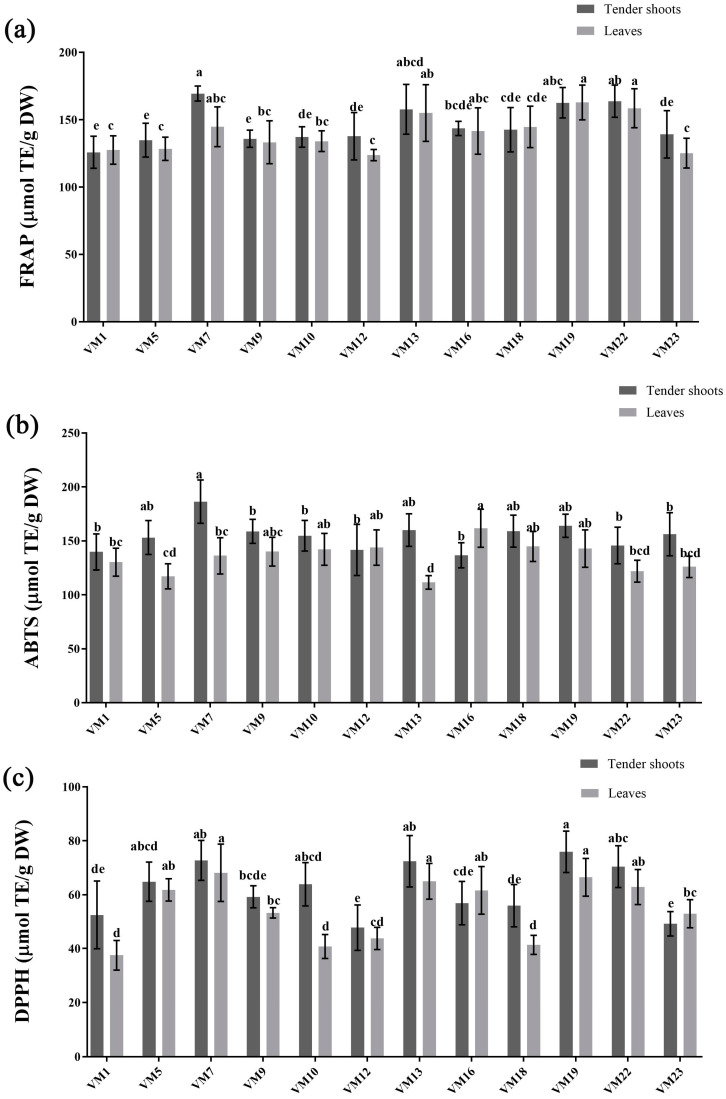
The total antioxidant activities of tender shoots and leaves in differential mulberry varieties. (**a**) FRAP values; (**b**) ABTS radical scavenging activities; (**c**) DPPH radical scavenging activities. The data are mean values of three replicates and standard deviation of the mean, and the different small letter superscripts within the same group (tender shoots or leaves) of histograms represent significant differences at *p* < 0.05 (ANOVA and LSD test).

**Figure 5 plants-12-03748-f005:**
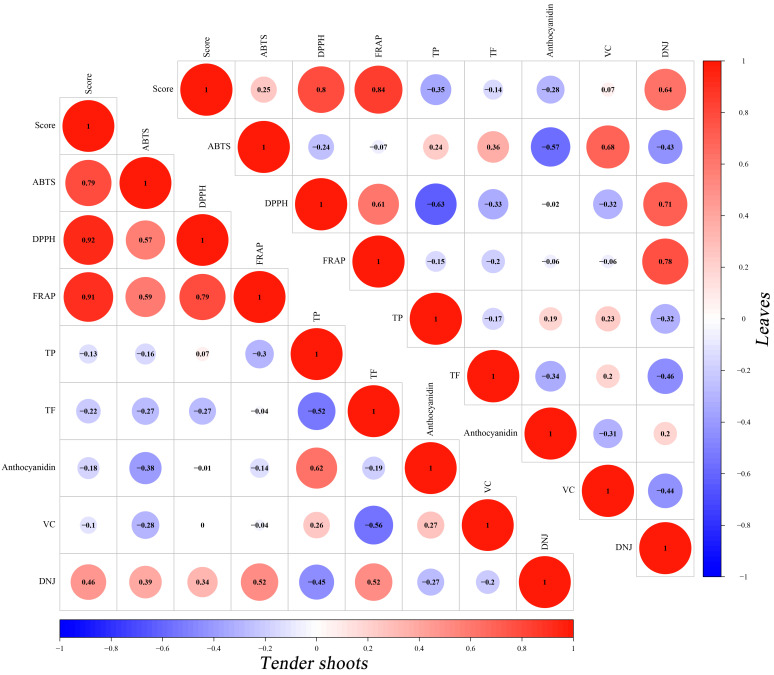
Pearson Correlation Coefficients between SP, carbohydrates, polyphenolic compounds, DNJ, VC, and the total antioxidant activities of tender shoots and leaves in differential mulberry varieties. Score: the comprehensive evaluation score of total antioxidant activities.

**Figure 6 plants-12-03748-f006:**
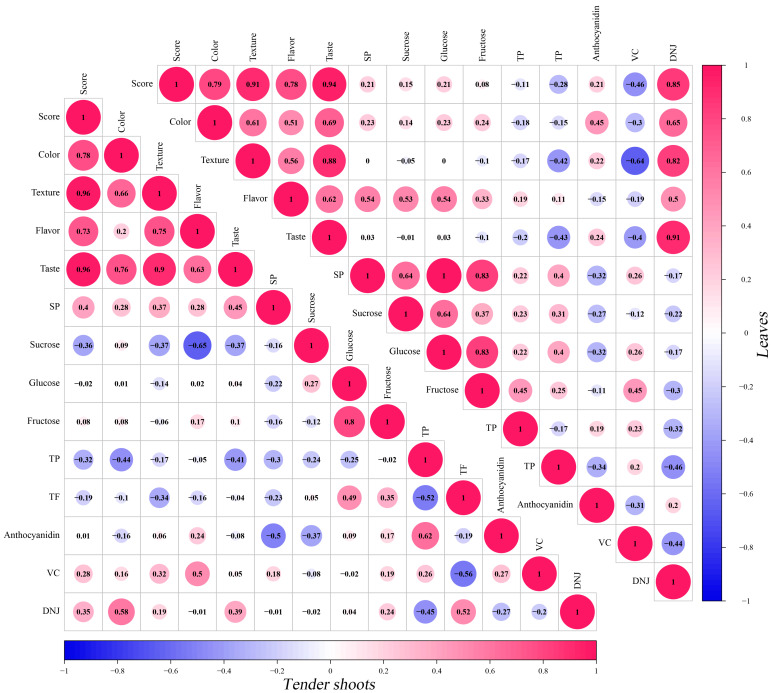
Pearson Correlation Coefficients between SP, carbohydrates, polyphenolic compounds, DNJ, VC, and sensory quality character indexes of tender shoots and leaves in differential mulberry varieties. Score: the comprehensive evaluation score (sensory).

**Table 1 plants-12-03748-t001:** The polyphenolic compound content in tender shoots and leaves of differential mulberry varieties.

Varieties	Anthocyanidin (mg of CG/g of DW)		TF (mg of QE/g of DW)		TP (mg of GAE/g of DW)	
	Tender Shoots	Leaves	Tender Shoots	Leaves	Tender Shoots	Leaves
*VM1*	3.19 ± 0.30 ^bc^	3.30 ± 0.29 ^ab^	15.58 ± 0.80 ^g^	15.96 ± 0.95 ^d^	48.01 ± 1.75 ^abc^	44.94 ± 1.88 ^ab^
*VM5*	2.93 ± 0.31 ^bc^	3.01 ± 0.26 ^bc^	19.400 ± 0.67 ^e^	16.10 ± 0.71 ^d^	48.21 ± 1.80 ^ab^	39.92 ± 1.72 ^e^
*VM7*	2.44 ± 0.16 ^c^	2.42 ± 0.11 ^de^	16.68 ± 0.71 ^fg^	13.89 ± 0.85 ^e^	45.78 ± 3.50 ^abcd^	40.21 ± 2.06 ^e^
*VM9*	3.14 ± 0.45 ^a^	2.22 ± 0.12 ^ef^	19.13 ± 0.38 ^e^	17.91 ± 0.43 ^c^	47.77 ± 0.49 ^abc^	44.37 ± 2.19 ^abcd^
*VM10*	3.49 ± 0.33 ^b^	3.38 ± 0.25 ^a^	17.65 ± 0.68 ^f^	15.85 ± 0.31 ^d^	49.38 ± 1.98 ^a^	46.80 ± 0.46 ^a^
*VM12*	2.91 ± 0.31 ^d^	2.66 ± 0.13 ^cd^	24.59 ± 1.01 ^a^	19.06 ± 0.76 ^ab^	45.26 ± 3.55 ^bcd^	42.79 ± 3.44 ^abcde^
*VM13*	2.33 ± 0.16 ^d^	3.19 ± 0.15 ^ab^	24.59 ± 0.82 ^a^	17.33 ± 0.77 ^c^	43.40 ± 0.39 ^d^	40.45 ± 1.95 ^de^
*VM16*	3.07 ± 0.39 ^bc^	2.33 ± 0.26 ^de^	20.94 ± 0.92 ^d^	17.45 ± 0.43 ^c^	48.70 ± 0.76 ^ab^	41.86 ± 2.93 ^bcde^
*VM18*	3.58 ± 0.24 ^a^	1.86 ± 0.09 ^f^	22.80 ± 0.80 ^c^	19.35 ± 0.41 ^a^	47.68 ± 3.35 ^abcd^	42.15 ± 1.81 ^bcde^
*VM19*	3.35 ± 0.23 ^b^	3.00 ± 0.12 ^bc^	15.52 ± 0.81 ^g^	18.37 ± 0.66 ^abc^	49.34 ± 1.83 ^a^	41.38 ± 1.60 ^bcde^
*VM22*	3.74 ± 0.04 ^a^	3.08 ± 0.30 ^ab^	19.90 ± 1.12 ^de^	13.62 ± 0.62 ^e^	45.66 ± 3.53 ^abcd^	44.61 ± 4.37 ^abc^
*VM23*	2.59 ± 0.19 ^d^	3.28 ± 0.28 ^ab^	19.03 ± 0.34 ^e^	18.03 ± 0.34 ^bc^	44.21 ± 1.82 ^cd^	40.85 ± 1.90 ^cde^
Mean	3.06	2.81	19.65	16.91	46.95	42.50
SD	0.45	0.498	3.13	1.86	2.01	2.20
CV (%)	14.56	17.72	15.93	11.01	4.29	5.16

The different small letter superscripts within the same column represent significant differences (*p* < 0.05) (ANOVA and LSD test). The data are mean values of three replicates and standard deviation of the mean.

**Table 2 plants-12-03748-t002:** Color, texture, flavor, taste, and sensory score of tender shoots and leaves in differential mulberry varieties.

Varieties	Tender Shoots						Leaves					
	Color	Texture	Flavor	Taste	Score	Ranking	Color	Texture	Flavor	Taste	Score	Ranking
*VM1*	4.67 ± 0.65 ^c^	4.42 ± 0.67 ^bc^	4.42 ± 0.79 ^b^	3.33 ± 0.49 ^d^	0.45 ± 0.14	7	4.42 ± 0.70 ^b^	2.67 ± 0.65 ^d^	4.42 ± 0.79 ^e^	2.67 ± 0.65 ^b^	0.42 ± 0.08	7
*VM5*	4.50 ± 0.67 ^c^	3.17 ± 0.39 ^d^	2.25 ± 0.45 ^d^	1.83 ± 0.39 ^f^	0.14 ± 0.29	12	4.58 ± 0.79 ^b^	2.17 ± 0.39 ^e^	2.25 ± 0.452 ^f^	0.83 ± 0.25 ^d^	0.20 ± 0.25	12
*VM7*	6.75 ± 0.75 ^ab^	6.75 ± 0.75 ^a^	4.67 ± 0.78 ^b^	6.50 ± 0.80 ^a^	0.89 ± 0.19	3	4.17 ± 0.72 ^b^	4.33 ± 0.65 ^b^	4.92 ± 070 ^cd^	6.33 ± 0.78 ^a^	0.70 ± 0.23	4
*VM9*	4.58 ± 0.67 ^c^	4.17 ± 0.58 ^dc^	4.08 ± 0.67 ^bc^	2.83 ± 0.39 ^e^	0.38 ± 0.17	10	2.25 ± 0.45 ^c^	1.83 ± 0.39 ^e^	4.67 ± 0.78 ^d^	1.25 ± 0.45 ^d^	0.20 ± 0.24	11
*VM10*	2.33 ± 0.49 ^d^	4.58 ± 0.52 ^bc^	4.50 ± 0.67 ^b^	4.08 ± 0.52 ^bc^	0.41 ± 0.16	8	4.67 ± 0.78 ^b^	1.50 ± 0.52 ^f^	4.83 ± 0.72 ^d^	1.83 ± 0.72 ^c^	0.34 ± 0.25	8
*VM12*	4.75 ± 0.45 ^c^	4.50 ± 0.52 ^bc^	4.25 ± 0.87 ^b^	4.33 ± 0.49 ^b^	0.50 ± 0.10	5	4.58 ± 0.67 ^b^	3.67 ± 0.65 ^c^	6.08 ± 0.79 ^bc^	2.00 ± 0.60 ^c^	0.56 ± 0.27	5
*VM13*	6.33 ± 0.65 ^b^	4.83 ± 0.58 ^bc^	4.67 ± 0.78 ^b^	6.17 ± 0.72 ^a^	0.72 ± 0.22	4	6.50 ± 0.80 ^a^	4.33 ± 0.65 ^b^	6.67 ± 0.49 ^ab^	6.50 ± 0.67 ^a^	0.93 ± 0.07	3
*VM16*	1.25 ± 0.45 ^e^	4.25 ± 0.45 ^c^	4.58 ± 1.00 ^b^	2.58 ± 0.70 ^e^	0.32 ± 0.36	11	4.67 ± 0.89 ^b^	1.67 ± 0.49 ^f^	3.75 ± 0.75 ^e^	1.08 ± 0.29 ^d^	0.26 ± 0.22	9
*VM18*	4.50 ± 0.67 ^c^	4.00 ± 0.74 ^d^	3.58 ± 0.79 ^c^	3.83 ± 0.58 ^c^	0.39 ± 0.14	9	4.75 ± 0.89 ^b^	1.17 ± 039 ^f^	6.62 ± 0.58 ^b^	1.83 ± 0.58 ^c^	0.42 ± 0.43	6
*VM19*	6.92 ± 0.67 ^a^	6.42 ± 0.67 ^bc^	5.75 ± 0.75 ^a^	6.33 ± 0.52 ^a^	0.93 ± 0.05	2	6.58 ± 0.79 ^a^	4.42 ± 0.67 ^b^	6.25 ± 0.97 ^bc^	6.33 ± 0.49 ^a^	0.91 ± 0.42	2
*VM22*	6.83 ± 0.72 ^ab^	6.67 ± 0.99 ^b^	6.25 ± 0.87 ^a^	6.58 ± 0.45 ^a^	0.99 ± 0.01	1	6.75 ± 1.06 ^a^	5.00 ± 0.60 ^a^	6.75 ± 0.87 ^a^	6.67 ± 0.49 ^a^	1.00 ± 0.00	1
*VM23*	4.42 ± 0.67 ^c^	4.33 ± 0.49 ^bc^	4.33 ± 0.78 ^b^	3.75 ± 0.45 ^cd^	0.45 ± 0.11	6	4.67 ± 0.89 ^b^	1.25 ± 0.45 ^f^	3.33 ± 0.49 ^f^	2.17 ± 0.58 ^c^	0.26 ± 0.21	10
Mean	4.82	4.84	4.53	4.35	0.55		4.88	2.83	5.05	3.29	0.52	
SD	1.75	1.14	1.05	1.66	0.27		1.24	1.42	1.46	2.39	0.30	
CV (%)	36.40	23.63	23.09	38.20	49.09		25.46	50.24	29.01	72.62	57.69	

The different small letter superscripts within the same column represent significant differences (*p* < 0.05) (ANOVA and LSD test). Data are mean values of sensory score of twelve food sensory evaluators and standard deviation. Score: the comprehensive evaluation score (sensory).

## Data Availability

The data presented in this study are available in this manuscript.

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
