# Peer review of "Preliminary Comparisons of Tender Shoots and Young Leaves of 12 Mulberry Varieties as Vegetables and Constituents Relevant for Their Potential Use as Functional Food for Blood Sugar Control"

_plants, 2023, doi:10.3390/plants12213748_

Round 1
Reviewer 1 Report
The overall idea of this study is to test both chemical constituents and sensory properties of shoots and leaves of a range of mulberry cultivars to identify the most suitable for use as vegetables. This is an original and relevant idea, however the study as described has multiple deficiencies, which must be alleviated before possible publication. Most likely the research would have to be repeated with an improved methodology to achieve a publishable level of scientific quality.
The Introduction needs more detail on the paper's topic, the use of mulberry shoots as a vegetable. Is this a traditional use or a new concept? Is it safe to eat for humans, and how much should be consumed to obtain the health benefits? Is there an advantage to consume it raw rather than cooked?
Reference 6 is completely irrelevant.
The antioxidant capacity of a food is not relevant for human health (see https://www.efsa.europa.eu/en/efsajournal/pub/5136), so it is irrelevant to measure it in the context of food quality.
Methods
The design is not properly described, specifically it is not clear how many field replicates were used. It says that samples were collected from 30 trees, but does not specify if this was 30 trees in total or 30 trees per cultivar. In either case it is not clear if the samples from each tree or each group of trees were kept as at least 3 separate field replications (correct approach) or pooled all in one sample from each cultivar (incorrect approach).
Looking at the unusually low SD values reported, it appears as if they were pooled, and that the 'triplicate' measurements were simply that each extract was analysed 3 times without any true replicates (in a true replicate, all the leaves must come from a tree or group of trees that is different from those used in the other replicates); if this was the case the SD then only reflects the purely analytical variability and does not give any information about the variation among trees within a cultivar.
Normal standard for publication of the results of a cultivar comparison is to test at least 3 replications within each cultivar, and to repeat this in at least 2 years, preferably 3 years (to control for the interaction with weather conditions). This will show which cultivars are best in the location where these plants were grown, and can be published, even though it will not show which cultivar should be recommended in general.
To test which cultivar is best overall, it is additionally necessary to include samples from different locations throughout the relevant area where this plant is cultivated. Normally this is done as a second step, selecting the 3-6 best cultivars to investigate based on the initial trial in one location.
For the sensory testing, it is not clear how the overall score was measured or calculated.
Hedonic scales ('like' versus 'dislike') can be justified when the purpose is only to identify the product that is most popular with consumers. However, these scales are not suitable for scientific use (such as correlation with other measures). For this purpose (as in the present study) the sensory properties should be measured using analytical scales (from 'very little' to 'very much'), of specific attributes such as sweetness, bitterness, aftertaste and so on, and preferably using an unstructured Visual Analogue Scale. This will provide a continuous value distribution much better suited for a correlation analysis than the discrete values from the 9-point scale.
The HPLC analyses are relevant, for ascorbic acid, 1-Deoxynojirimycin and the sugars, and it is acceptable to use the recognised colorimetric method for protein determination and the relatively direct measurement of anthocyanin.
In contrast, the colorimetric assays for flavonoids and for phenolics are far too sensitive to interference from other compounds to be useful in the context. For example, the Folin reagent also reacts with protein (it is even used for this purpose in the Lowry method).
Results
The data presentation is very deficient.
For example, Figure 1b shows no units, consists mainly of abbreviations that are not explained and does not show any variance. The only way to make sense of this table is to look at the table in the Supplementary material! Table 1 (this also applies to Table 2) is clear but all the values are shown with too many significant digits; 2 or 3 digits is plenty, which means that most of the values should show only one digit after the decimal point, and round off the other two.
Also the coefficient of variation (CV) among the cultivars is not of scientific interest, it just reflects the SD. What is important is the average of the CVs within cultivars at each level: CV for variation among replications (e.g. trees), and CV for variation among repeated measures of the same extract. CV for replications (which seems to be missing) shows how variable the products are, while the CV for the repetitions show how precise the analysis is.
For vegetables for human consumption, the relevant values should be presented per 100g of fresh weight (in the same way as nutrient contents are always presented in food databases), not per g DW. Also the DW% of each analysed material in one of the table; this is an important result in its own right. Additionally, this will allow the reader to calculate the composition on a dry matter basis if they want to use the information in one of the few contexts (for example animal feed use) where this format is more useful than the fresh weight basis.
In Figure 2 the statistics are wrong, since the data on leaves are analysed in the same comparison as the data on shoots. Also the letters are too small, otherwise the presentation of this figure is OK.
In Figure 3, the statistics in panes a-c are OK (the leves are compared only with leaves, and shoots only with shoots), but the format is wrong; data points in a figure should ONLY be connected if they show the SAME sample measured at different time points (for example, this format would be appropriate to show values from the same tree measured in different months during a year).
The format of panel d in Figure 3 is fine (except that the text is too small), however the interpretation in the figure legend is incorrect: there is no evidence of an 'influence' of DNJ on the antioxidant measures, only an 'association'. Which could be caused by many different influences, including all those that were not measured in the study.
The description of the sensory colour scores in lines 310-313 does not correspond to using a hedonic scale (like/dislike), it would be the results of an analytical scale (such as light/dark). It is not clear if this is an error in the method description or in the results, but at least one of them are wrong.
Figure 4 is too crowded, almost impossible to read the text. The colour coding is useful, and the results would have been impressive if they had been based on samples with true replicates and appropriately measurements.
Discussion
Where possible (e.g. refs 31-34) compare with literature on tree leaves/shoots rather than vegetables.
The discussion regarding which cultivars are best, should be based on references to relevant well-designed placebo-controlled human intervention trials. If such studies have not been published, the discussion should focus on how the present study could be useful for researchers to carry out such a trial, but it should not claim any health benefits.
Overall, the present data could be useful as pilot data for the design of a properly replicated study, which could then in time result in publishable results. However without the replications, the results shown in the present manuscript are completely unreliable. They cannot be used to make any conclusions, other than indicating that there is sufficient variation among cultivars to justify doing a well-designed study to investigate them appropriately.
Most of the manuscript is written in good English.
There are a few sentences with grammatical errors or non-optimal words, e.g. lines 93-103 and lines 147-148.
Author Response
Comments and Suggestions for Authors
The overall idea of this study is to test both chemical constituents and sensory properties of shoots and leaves of a range of mulberry cultivars to identify the most suitable for use as vegetables. This is an original and relevant idea, however the study as described has multiple deficiencies, which must be alleviated before possible publication. Most likely the research would have to be repeated with an improved methodology to achieve a publishable level of scientific quality.
The Introduction needs more detail on the paper's topic, the use of mulberry shoots as a vegetable. Is this a traditional use or a new concept? Is it safe to eat for humans, and how much should be consumed to obtain the health benefits? Is there an advantage to consume it raw rather than cooked?
Answer:In response to the reviewer's inquiries, we have added details in the introduction to clarify that the use of mulberry shoots as a vegetable is a traditional practice with a long history. We also emphasized the safety of mulberry leaves for human consumption, as supported by research papers and recognition by the Chinese Ministry of Health. Regarding the optimal consumption for health benefits, much of the research on the health benefits of mulberry leaves has focused on their potential to lower blood glucose levels. Studies have shown that consuming mulberry leaf powder, such as 3g dissolved in water alongside biscuits, significantly reduces postprandial glucose levels in young adults with elevated glucose levels(New reference 1). As for raw versus cooked consumption, one of the most studied bioactive compounds in mulberry leaves, 1-DNJ, is heat-stable, meaning that its activity is not significantly affected by cooking(New references 6). However, it's important to note that certain heat-sensitive compounds, such as vitamin C, may degrade when exposed to high temperatures. Therefore, whether mulberry leaves are consumed raw or cooked may depend on the specific compounds of interest, and further research may be needed to determine the advantages of one method over the other.
Reference 6 is completely irrelevant.
Answer:Thank you for pointing out the issue with Reference 6. We have removed this irrelevant reference from our paper and added new relevant reference.
The antioxidant capacity of a food is not relevant for human health (see https://www.efsa.europa.eu/en/efsajournal/pub/5136), so it is irrelevant to measure it in the context of food quality.
Answer:
In response to your question about the relevance of antioxidant capacity in food to human health, we believe that while there is ongoing debate, it is not widely accepted that antioxidant capacity is entirely unrelated to human health. Additionally, it's worth noting that antioxidant capacity is crucial for the quality of the food itself(New reference 9). In fact, it remains an important parameter in many food and nutrition studies. Furthermore, consumers are quite interested in the antioxidant content of the foods they consume, making it a relevant factor in food quality and your choices as a consumer. Furthermore, numerous studies have shown that the consumption of high-antioxidant foods, particularly fruits, can lead to a temporary increase in the total antioxidant capacity (TAC) of blood plasma shortly after ingestion, suggesting a potential connection between antioxidant capacity and physiological responses(New references 7,New references 8). Thus, although the relationship between antioxidant capacity and human health is subject to debate, it is premature to conclude that it is entirely irrelevant, as it continues to be a significant parameter in the fields of food science and nutrition.
Methods
The design is not properly described, specifically it is not clear how many field replicates were used. It says that samples were collected from 30 trees, but does not specify if this was 30 trees in total or 30 trees per cultivar. In either case it is not clear if the samples from each tree or each group of trees were kept as at least 3 separate field replications (correct approach) or pooled all in one sample from each cultivar (incorrect approach).
Answer:We have modified the description about the samples collection, and the samples were collected from 30 trees per cultivar.
Looking at the unusually low SD values reported, it appears as if they were pooled, and that the 'triplicate' measurements were simply that each extract was analysed 3 times without any true replicates (in a true replicate, all the leaves must come from a tree or group of trees that is different from those used in the other replicates); if this was the case the SD then only reflects the purely analytical variability and does not give any information about the variation among trees within a cultivar.
Answer:In future, we will conduct indicator measurement and data analysis about the mulberry tree varieties we have chosen for different locations based on the opinions of participating reviewers.
Normal standard for publication of the results of a cultivar comparison is to test at least 3 replications within each cultivar, and to repeat this in at least 2 years, preferably 3 years (to control for the interaction with weather conditions). This will show which cultivars are best in the location where these plants were grown, and can be published, even though it will not show which cultivar should be recommended in general.
Answer:We are conducting a preliminary screening of suitable mulberry tree varieties for consumption. Therefore, we have not yet conducted sample collection for many years. In future, we will conduct indicator measurement and data analysis for many years based on the opinions of participating reviewers.
To test which cultivar is best overall, it is additionally necessary to include samples from different locations throughout the relevant area where this plant is cultivated. Normally this is done as a second step, selecting the 3-6 best cultivars to investigate based on the initial trial in one location.
Answer:We are conducting a preliminary screening of suitable mulberry tree varieties for consumption. Therefore, we have not yet conducted sample collection for different locations. In future, we will conduct indicator measurement and data analysis about the mulberry tree varieties we have chosen for different locations based on the opinions of participating reviewers.
For the sensory testing, it is not clear how the overall score was measured or calculated.
Answer:We have modified the description about the sensory evaluation methods, especially the mensuration and calculation about the overall score of sensory evaluation.
Hedonic scales ('like' versus 'dislike') can be justified when the purpose is only to identify the product that is most popular with consumers. However, these scales are not suitable for scientific use (such as correlation with other measures). For this purpose (as in the present study) the sensory properties should be measured using analytical scales (from 'very little' to 'very much'), of specific attributes such as sweetness, bitterness, aftertaste and so on, and preferably using an unstructured Visual Analogue Scale. This will provide a continuous value distribution much better suited for a correlation analysis than the discrete values from the 9-point scale.
Answer:We have modified the description about the sensory evaluation methods, and added the borrowed sensory evaluation standard of mulberry leaf vegetable form a previous report.
The HPLC analyses are relevant, for ascorbic acid, 1-Deoxynojirimycin and the sugars, and it is acceptable to use the recognised colorimetric method for protein determination and the relatively direct measurement of anthocyanin. In contrast, the colorimetric assays for flavonoids and for phenolics are far too sensitive to interference from other compounds to be useful in the context. For example, the Folin reagent also reacts with protein (it is even used for this purpose in the Lowry method).
Answer:For the determination of flavonoids, polyphenols, and proteins, we conducted them according to existing standards and did not use a more precise method of liquid equivalence. In this study, although some samples were studied using parallel methods, the differences in content between varieties were compared.
Results
The data presentation is very deficient.
For example, Figure 1b shows no units, consists mainly of abbreviations that are not explained and does not show any variance. The only way to make sense of this table is to look at the table in the Supplementary material! Table 1 (this also applies to Table 2) is clear but all the values are shown with too many significant digits; 2 or 3 digits is plenty, which means that most of the values should show only one digit after the decimal point, and round off the other two.
Answer:We have separated Fig. 1 in old version of manuscript as Fig. 1 and Fig. 2 in modified manuscript, and modified the figure, and added units and abbreviations SP, S, G, F mean in Figure. Also, we have modified the data in all manuscript and the Supplementary material, and show two digits after the decimal point.
Also the coefficient of variation (CV) among the cultivars is not of scientific interest, it just reflects the SD. What is important is the average of the CVs within cultivars at each level: CV for variation among replications (e.g. trees), and CV for variation among repeated measures of the same extract. CV for replications (which seems to be missing) shows how variable the products are, while the CV for the repetitions show how precise the analysis is.
Answer:We will calculate the CV for variation among replications (e.g. trees), and CV for variation among repeated measures of the same extract in many years and many location data in future.
For vegetables for human consumption, the relevant values should be presented per 100g of fresh weight (in the same way as nutrient contents are always presented in food databases), not per g DW. Also the DW% of each analysed material in one of the table; this is an important result in its own right. Additionally, this will allow the reader to calculate the composition on a dry matter basis if they want to use the information in one of the few contexts (for example animal feed use) where this format is more useful than the fresh weight basis.
Answer:The main purpose of this study is to compare the material differences between varieties, and the data representation per g DW methods used in most literature refer to methods.
In Figure 2 the statistics are wrong, since the data on leaves are analysed in the same comparison as the data on shoots. Also the letters are too small, otherwise the presentation of this figure is OK.
Answer:In this study, the index between shoot leaves and leaves are analyzed, for comparing the differences between shoot leaves and leaves. We have enlarged the figure.
In Figure 3, the statistics in panes a-c are OK (the leaves are compared only with leaves, and shoots only with shoots), but the format is wrong; data points in a figure should ONLY be connected if they show the SAME sample measured at different time points (for example, this format would be appropriate to show values from the same tree measured in different months during a year).
Answer:We have separated Fig. 3 in old version of manuscript as Fig. 4 and Fig. 5 in modified manuscript, and modified the figure, explained what the abbreviations SP, S, G, F mean and add "protein" to "sugar profile" in the title of the figure.
The format of panel d in Figure 3 is fine (except that the text is too small), however the interpretation in the figure legend is incorrect: there is no evidence of an 'influence' of DNJ on the antioxidant measures, only an 'association'. Which could be caused by many different influences, including all those that were not measured in the study.
Answer: We have separated Fig. 3d in old version of manuscript as Fig. 5 in modified manuscript, and modified the figure legend. Additionally, changed the 'influence' as 'association' in all manuscript.
The description of the sensory colour scores in lines 310-313 does not correspond to using a hedonic scale (like/dislike), it would be the results of an analytical scale (such as light/dark). It is not clear if this is an error in the method description or in the results, but at least one of them are wrong.
Answer: We have added in modified sensory evaluations method and standard in modified manuscript in ‘Method’section, and modified the description of the sensory color, texture, flavor and taste scores in lines 310-313.
Figure 4 is too crowded, almost impossible to read the text. The colour coding is useful, and the results would have been impressive if they had been based on samples with true replicates and appropriately measurements.
Answer: We have changed the Figure 4 in modified manuscript follow the suggestion from reviewer.
Discussion
Where possible (e.g. refs 31-34) compare with literature on tree leaves/shoots rather than vegetables.
The discussion regarding which cultivars are best, should be based on references to relevant well-designed placebo-controlled human intervention trials. If such studies have not been published, the discussion should focus on how the present study could be useful for researchers to carry out such a trial, but it should not claim any health benefits.
Overall, the present data could be useful as pilot data for the design of a properly replicated study, which could then in time result in publishable results. However without the replications, the results shown in the present manuscript are completely unreliable. They cannot be used to make any conclusions, other than indicating that there is sufficient variation among cultivars to justify doing a well-designed study to investigate them appropriately.
Answer: We have delved into a more comprehensive and in-depth discussion of the data in modified manuscript according to suggestion from all reviewers, and hope to give existing dependencies for all reviewers.
Comments on the Quality of English Language
Most of the manuscript is written in good English.
There are a few sentences with grammatical errors or non-optimal words, e.g. lines 93-103 and lines 147-148.
Answer: The error has been modified, and we also carefully checked the manuscript, and have found and modified errors in all manuscript.

Reviewer 2 Report
The primary aim of this study is to investigate and compare the nutritional composition and 1-Deoxynojirimycin (DNJ) content within a selection of twelve distinct mulberry tree cultivars, with a specific focus on their tender shoots and leaves.
Mulberry species, along with their various cultivars and plant organs, particularly leaves, have undergone scrutiny about their nutritional profiles and various biological attributes, encompassing antioxidant, antibacterial, antiproliferative, anticancer, and antihyperglycemic properties, as well as the compounds responsible for these effects. Given the existing knowledge base concerning this botanical specimen, the novelty of the present research is relatively limited, since the study is centered on the characterization and comparison of properties of mulberry cultivars. This study primarily serves as a description of variances, whether increase or decrease, in parameters across the different cultivars.
Additionally, it is anticipated that correlations, specifically the positive associations between antioxidant activity and the methodologies (i.e., ABTS, FARP, and DPPH) employed for its assessment, would manifest as anticipated outcomes. Moreover, the higher or lower correlation between antioxidant activity and the methods is related to the chemical principles of the analytical methods.
Furthermore, discrepancies in secondary metabolite and sugar content observed between tender shoots and leaves can be attributed to the inherent physiological characteristics of the organs of plant species in general.
Moreover, the authors suggest that when quality indicators are combined with sensory evaluations, mulberry tree varieties exhibiting high DNJ content exhibit higher antioxidant activity and receive higher scores in sensory evaluations, suggesting that DNJ content can serve as a reference indicator when selecting suitable mulberry tree varieties for consumption. However, it is important to note that this conclusion carries some risk, as the sensory evaluation involved only 12 evaluators, despite their good experience in vegetable sensory evaluation. Furthermore, considering the range of values of sugars and DNJ present in the different varieties, the relationship between the range of sugars and DNJ content found in the various tender shoots remains somewhat unclear. For instance, it is not clear why tender shoot VM7 (with DNJ = 7.332 mg /gDW) may be more suitable for individuals seeking to manage blood sugar levels than tender shoot V16 (with DNJ = 6.240 mg/gDW) and with much lower content of sucrose and similar or levels of reducing sugars. Does the difference of 1 mg/g DW in tender shoot have an impact on human health?
The authors should take into account the previously mentioned aspects and delve into a more comprehensive and in-depth discussion of the data.
Minor comments:
Line 297: The ABTS, FRAP and DPPH are methods and not activities.
Line 112: please, refer the time in extraction of flavonoids.
Line 134: regarding anthocyanin, please explain why the values are not expressed per gram of extracts, as in the other methods.
Please, refer the yield of the extractions.
Author Response
Comments and Suggestions for Authors
The primary aim of this study is to investigate and compare the nutritional composition and 1-Deoxynojirimycin (DNJ) content within a selection of twelve distinct mulberry tree cultivars, with a specific focus on their tender shoots and leaves.
Mulberry species, along with their various cultivars and plant organs, particularly leaves, have undergone scrutiny about their nutritional profiles and various biological attributes, encompassing antioxidant, antibacterial, antiproliferative, anticancer, and antihyperglycemic properties, as well as the compounds responsible for these effects. Given the existing knowledge base concerning this botanical specimen, the novelty of the present research is relatively limited, since the study is centered on the characterization and comparison of properties of mulberry cultivars. This study primarily serves as a description of variances, whether increase or decrease, in parameters across the different cultivars.
Additionally, it is anticipated that correlations, specifically the positive associations between antioxidant activity and the methodologies (i.e., ABTS, FARP, and DPPH) employed for its assessment, would manifest as anticipated outcomes. Moreover, the higher or lower correlation between antioxidant activity and the methods is related to the chemical principles of the analytical methods.
Furthermore, discrepancies in secondary metabolite and sugar content observed between tender shoots and leaves can be attributed to the inherent physiological characteristics of the organs of plant species in general.
Moreover, the authors suggest that when quality indicators are combined with sensory evaluations, mulberry tree varieties exhibiting high DNJ content exhibit higher antioxidant activity and receive higher scores in sensory evaluations, suggesting that DNJ content can serve as a reference indicator when selecting suitable mulberry tree varieties for consumption. However, it is important to note that this conclusion carries some risk, as the sensory evaluation involved only 12 evaluators, despite their good experience in vegetable sensory evaluation. Furthermore, considering the range of values of sugars and DNJ present in the different varieties, the relationship between the range of sugars and DNJ content found in the various tender shoots remains somewhat unclear. For instance, it is not clear why tender shoot VM7 (with DNJ = 7.332 mg /gDW) may be more suitable for individuals seeking to manage blood sugar levels than tender shoot V16 (with DNJ = 6.240 mg/gDW) and with much lower content of sucrose and similar or levels of reducing sugars. Does the difference of 1 mg/g DW in tender shoot have an impact on human health?
The authors should take into account the previously mentioned aspects and delve into a more comprehensive and in-depth discussion of the data.
Answer: We have delved into a more comprehensive and in-depth discussion of the data in modified manuscript according to suggestion from reviewer.
Minor comments:
Line 297: The ABTS, FRAP and DPPH are methods and not activities.
Answer: Change the model representation of the ABTS, FRAP and DPPH in modified manuscript.
Line 112: please, refer the time in extraction of flavonoids.
Answer: Added the time in extraction of flavonoids in modified manuscript.
Line 134: regarding anthocyanin, please explain why the values are not expressed per gram of extracts, as in the other methods.
Answer: Changed the anthocyanin values unit, and expressed per gram of extracts in modified manuscript as in the other methods.
Please, refer the yield of the extractions.
Answer: We need to count the yield of the extractions in future.

Reviewer 3 Report
The manuscript is interesting and contains a lot of valuable information, but I found a few elements that require correction or clarification. The aim of the study was to compare the nutritional composition of tender shoots and leaves of twelve mulberry varieties.
The first note concerns the markings. Authors often mix abbreviations and full names of various indicators. When using it for the first time, write the full name and the abbreviation in brackets, and then use only the abbreviations, as is the case with DNJ and VC. This applies, for example, to lines 230-234, 289-291 and others.
It is completely incomprehensible, to combine the photo and the diagram marked in Figure 1. Charts or drawings can be combined with each other if they concern similar features. In this case, Fig. 1 should be separated as Fig. 1 and Fig. 2. Moreover, it should be explained what the abbreviations SP, S, G, F mean and add "protein" to "sugar profile" in the title of the figure.
A similar remark applies to Figure 3. The graphs (as Figure 4 a,b,c) should be separated from the table with the Pearson correlation indices (as Figure 5).
Lines 243-244, 262-263, 273-274, 282-283 - these sentences should be included in the ‘Discussion’ section.
Line 257 – the table shows differently "from 43.399 to 49.382"
Table 1 - in the second column "Anthocyanidin - Tender shoots", homogeneous groups are incorrectly marked (e.g. 3.73a, 2.59a, 3.14ab...). There are also errors in the average values. Please check everything carefully and correct it. Additionally, there is a lack of explanation of the abbreviations SD and VC.
Line 289-290 – Was DPPH actually higher than ABTS and FRAP? The scale on the x-axis says something completely different.
Lines 297-307 – the Pearson Correlation indicators and the entire description should be carefully analyzed again. For me this description is incomprehensible in some places.
Line 313, 316, 317 and 319 – the numerical values in brackets do not correspond to the values in table 2.
The discussion is short and rather superficial. I suggest expanding it further to explain the existing dependencies.
Please separate ‘Conclusion’ as a separate subsection.
Author Response
Comments and Suggestions for Authors
The manuscript is interesting and contains a lot of valuable information, but I found a few elements that require correction or clarification. The aim of the study was to compare the nutritional composition of tender shoots and leaves of twelve mulberry varieties.
The first note concerns the markings. Authors often mix abbreviations and full names of various indicators. When using it for the first time, write the full name and the abbreviation in brackets, and then use only the abbreviations, as is the case with DNJ and VC. This applies, for example, to lines 230-234, 289-291 and others.
Answer: We have carefully modified the mix abbreviations and full names of various indicators in modified manuscript.
It is completely incomprehensible, to combine the photo and the diagram marked in Figure 1. Charts or drawings can be combined with each other if they concern similar features. In this case, Fig. 1 should be separated as Fig. 1 and Fig. 2. Moreover, it should be explained what the abbreviations SP, S, G, F mean and add "protein" to "sugar profile" in the title of the figure.
Answer: We have separated Fig. 1 in old version of manuscript as Fig. 1 and Fig. 2 in modified manuscript, and modified the figure, explained what the abbreviations SP, S, G, F mean and add "protein" to "sugar profile" in the title of the figure.
A similar remark applies to Figure 3. The graphs (as Figure 4 a,b,c) should be separated from the table with the Pearson correlation indices (as Figure 5).
Answer: We have separated Fig. 3 in old version of manuscript as Fig. 4 and Fig. 5 in modified manuscript.
Lines 243-244, 262-263, 273-274, 282-283 - these sentences should be included in the ‘Discussion’ section.
Answer: These sentences of Lines 243-244, 262-263, 273-274, 282-283 have been moved to ‘Discussion’ section.
Line 257 – the table shows differently "from 43.399 to 49.382"
Answer: The error has been modified, and we also carefully checked the manuscript, and have found and modified errors in all manuscript.
Table 1 - in the second column "Anthocyanidin - Tender shoots", homogeneous groups are incorrectly marked (e.g. 3.73a, 2.59a, 3.14ab...). There are also errors in the average values. Please check everything carefully and correct it. Additionally, there is a lack of explanation of the abbreviations SD and VC.
Answer: These errors has been modified, and we also carefully found and modified errors in all manuscript, and added the explanation of the abbreviations SD and VC.
Line 289-290 – Was DPPH actually higher than ABTS and FRAP? The scale on the x-axis says something completely different.
Answer: We have changed the model representation, there are the differences between variers (coefficient of variation) about DPPH, ABTS and FRAP, not the activities about DPPH, ABTS and FRAP.
Lines 297-307 – the Pearson Correlation indicators and the entire description should be carefully analyzed again. For me this description is incomprehensible in some places.
Answer: We have carefully analyzed the Pearson Correlation indicators and the entire description again in the modified manuscript.
Line 313, 316, 317 and 319 – the numerical values in brackets do not correspond to the values in table 2.
Answer: These errors has been modified, and we also carefully found and modified errors in all manuscript.
The discussion is short and rather superficial. I suggest expanding it further to explain the existing dependencies.
Answer: We have delved into a more comprehensive and in-depth discussion of the data in modified manuscript according to suggestion from all reviewers, and hope to give existing dependencies for all reviewers.
Please separate ‘Conclusion’ as a separate subsection.
Answer: We have separated ‘Conclusion’ as a separate subsection in the modified manuscript.

Round 2
Reviewer 1 Report
The manuscript is very much improved, and it is appreciated that the authors express understanding of some of the limitations of the study's design. However, additional changes are still necessary before it becomes suitable for publication.
It is essential that the limitations of the present study (of being unreplicated and not including human trials other than sensory evaluation) must be more clearly shown in the manuscript, not only in the response to reviewers.
The title must be changed accordingly; for example, an acceptable title would be: 'Preliminary Comparisons of Tender Shoots and Young Leaves of 12 Mulberry Varieties as Vegetables and Constituents Relevant for their Potential use as Functional Food for Blood Sugar Control'. This title describes the data that are actually in the manuscript, not only speculations about what this knowledge could be used for, while still indicating the key aspect. The word 'Preliminary' may be replaced with another word with similar meaning, like 'Pilot' or 'Exploratory'.
Abstract
In line 20 delete this text: 'found to be the primary contributor to mulberry leaf antioxidant activity and', because the design does not allow determination of which compound(s) contributed to the antioxidant activity, only which one was most strongly associated.
Between lines 22 and 23 (after 'for' and before 'blood'), insert 'development as a functional food for', because much more research is needed before a functional food could be registered. Also, it is not clear why a higher sugar content would be superior for blood sugar control, since products for blood sugar control are usually chosen for the lowest sugar content. It might still be true, but this requires a human intervention trial to find out.
In line 25, delete 'individuals making informed' and replace it with 'future research into', because none of these varieties have been tested for blood sugar control in an appropriately designed and replicated trial.
Introduction
The question of antioxidant effect is not about whether researchers and the public believe in the health benefit, it is about what is good or bad science. The paper
In contrast, the introduction should provide more details on the studies reported in the recent papers on mulberry shoots and blood sugar control that are referenced in the introduction, instead of the antioxidant information. This work is recent and interesting, it has not been disproved and is directly relevant for the manuscript. There is no maximum limit to the length of the manuscript, so it would be fine to use a paragraph or two on this to educate the readers. However, it is also important to clearly specify (see https://www.gov.uk/government/publications/uknhcc-scientific-opinion-white-mulberry-leaf-extract-and-blood-glucose-levels/scientific-opinion-for-the-substantiation-of-a-health-claim-on-a-single-component-of-morus-alba-white-mulberry-leaf-extract-and-assisting-healthy-bl) that the evidence for the benefit of mulberry leaves is still not sufficiently firm to allow its use as a functional food in Europe. The regulatory rejection is not a disproval, only a request for more and better research data; but the results in the present manuscript can only be implemented globally after a sufficient volume of such additional research has been completed and demonstrated a consistent effect.
doi: 10.1007/s00394-007-0687-2. For examples of plant constituents with well-documented human health effects, better choices would be other vegetables/herbs such as garlic, ginseng, onion etc.
doi: 10.3389/fnut.2022.849841 (which also gives useful information about what is required before a food can be approved as 'functional' in Europe) and the clear outcome of this small animal trial:References 5 and 6 are not about anti-inflammatory effects of flavonoids, please replace with relevant references or remove/revise this sentence completely. Those references do appear to be relevant for the description of research on DNJ so may be used in that context instead.
Lines 83-84: EITHER remove the text 'not only offer individuals with high blood sugar levels the opportunity to make informed choices regarding their vegetable selection, but also' OR replace it with some mention of the potential for future approval as functional foods, WITHOUT giving the impression that your best performing variety is ready for this type of use. The 'informed choices' by 'individuals' is wrong, BOTH because of the lack of replications in the present trial, AND because of the still not sufficient human trial evidence. Both of these deficiencies can be alleviated by future research (and hopefully will do this soon), so this change does not detract from the manuscript, as long as the manuscript avoids making unsubstantiated promises.
Methods
Overall organisation: The methods corresponding to each of the results should be shown in the same sequence; at present, the sensory testing comes before phenolics in the Methods, but after in the Results. Please re-arrange either the Methods section or the Results section, so the sequecne of the subsections become consistent with each other. This also applies to the Discussion section, although it is not as important there.
Regarding sample preparation: Some of the samples were freeze dried (on pages 3, 4 and 5); include a small section with the details of the freeze drying method, in particular the equipment used and the maximum temperature reached during the process.
The sensory testing is now much more clearly described, in lines 115-118 specifying the analytical scales actually used, this is fine. But why does it still mention a hedonic scale in lines 109-110? Maybe a 'Likert scale' (with numbered points and a text equivalent for each point) was mislabelled as a 'hedonic scale' (with like/dislike text)? This is a common error, because Likert scales are often used with hedonic text. An analytical scale is a scale where the panellists are asked to assess the properties of a product, while a hedonic scale assessess the panelist's liking of the product. Check the literature for references on 'Likert scales' and 'Hedonic scales' and correct the text if relevant. If a hedonic scale was also used, then include the results in Table 2 as 'Preference'.
In lines 118-119 it says that the average scores were calculated, which is fine; however in Table 2, the values for 'Overall score' are not the average of the other scores. Find out where the error is and correct it.
In section 2.3 (the colorimetric analyses), there is only a reference for one of the 4 methods shown. The methods are defined by their names, which would be OK if they were done using so well-established and previously validated methods that no reference was necessary. However, the anthocyanin method described is very different from the established standard method with the same name (AOAC 2005.02: Total Monomeric Anthocyanin Pigment Content of Fruit Juices, Beverages, Natural Colorants, and Wines- pH Differential Method). EITHER provide the full reference to the article showing the validation and details of the method, OR consider to drop any mention of anthocyanins, if the reported values are not comparable to values measured using validated methods.
In line 163, the word 'properly' is not clear; check if there is a typo or other error? The content of the sentence is clear and appropriate, this just looks like a language error.
The DPPH assay also seems to lack a reference.
The box plots mentioned in line 228 are missing from the results. The word 'Evaluations' at the end of this line makes no sense; should it be 'Correlations'?
Results
Even though it is clearly stated in line 224 that all values are shown as Mean±SD, then this must be stated again in every table or figure legend.
If a table (like Table 2)or figure (like Figure 3) uses letters to present significant differences, then this must also be explained (in a few words) in the table heading or a footnote or figure legend. This explanation MUST specify if the significance was calculated separately for shoots and for leaves (which is correct), or with all these data mixed together (which would be incorrect) AND which method was used to control the type 1 error (for example the Tukey test).
As previously mentioned, it is best (albeit not essential) to report the concentrations of the constituents per 100g fresh weight. It is acceptable to report some of them (like the antioxidant measures) as µmol/g DW, since this is common in other papers.
The anthocyanin values in Table 1 are surprisingly high, and the differences do not match the colours shown on Figure 1. For example, the paper https://doi.org/10.3389/fpls.2023.1155722 showed that lettuce cultivars with red-coloured leaves contained between 0.2 and 1 mg/g DW anthocyanin. The leaves on Figure 1 are only slightly mauve in colour, much lighter than red lettuce leaves, so would be expected to contain less that 0.2 mg/g DW, according to this article. As mentioned before, the corresponding method description is also rather strange, so it is possible that the values are simply not correct. Since it is clear from the photos that the anthocyanin content is very low, the manuscript will not lose any important information if all mentions of anthocyanins are removed; this is better than including values that are not measured with a correct method.
For vitamin C, which was measured on fresh plant material, it is absurd to report the value as mg/100g DW; hopefully this was just a typo, and it actually means mg/100g FW! if so please correct the typo.
If it is not a typo, and the authors explicitly want to use only DW values for the reporting, then do it as mg/g DW, so at least it is comparable with the other values in the present study.
Note that if the mg/100g DW is NOT a typo, then the values for vitamin C are low (compared with other vegetables), and it is then NOT a rich source. In this case, correct the statement in line 295.
In any case, no matter which units are used, it is absolutely essential to report the Dry Matter percentage (gDM/100g FW) for each of the 24 samples, in one of the tables; with this information, the reader can compare the values in the present study with values from the literature, irrespective of which unit it is reported in.
The term 'score' is used both for antioxidant activity and for overall sensory evaluation. Make sure it is clear in every case which type of score it is about; in particular in Figure 5 and 6 and their figure legends.
Line 312: the text is clear but not correct English, please improve.
In Table 2, the SD, Mean, SD between varieties and CV between varieties are missing for the Overall score. As mentioned above, the values of the Overall score also do not match the mean of the other scores, as they should according to the method description. Please find out what the real values are, fix either the values or the method description so they match, and show all of the values; or, if they are an error that cannot be fixed, remove them so they don't confuse the reader.
The correlations between VC and DNJ are shown on both Figures 5 and 6, which is fine; but why is the value for the tender shoots different in the two figures (0.16 in Figure 5 and -0.44 in Figure 6)? For the leaves, the value is the same (-0.20) in both figures.
The text in lines 331-335 does not match the values shown on Figure 5! In particular in the tender shoots, the DNJ values on the figure are not correlated with anything, while the text says that the correlation is very high (0.63). Which is correct?
Please check that there are no errors in all the data in all the tables and figures! If the values are correct, then explain why the two figures gave different results, for example if there are different numbers of samples in separate calculations, and if so, what they are.
Discussion
Lines 412-413: Delete this sentence, for the reasons explained in the introduction. The authors of those references know much less about this topic than the experts at EFSA.
Line 426-27: Remove the word 'alarmingly', it is unnecessary, unsubstantiated and confusing when not explained in context.
Line 435: Remove the word 'vast', it is unnecessary, unsubstantiated and confusing when not explained in context. Also correct the grammar (singular/plural) of this sentence.
Line 437: the text is clear but not correct English, please improve.
Lines 443-444: Figure 5 shows little or no correlations between DNJ and antioxidant activities, despite stating the opposite in the text. Once the errors in the data have been found and corrected, so the results have changed, then the discussion must be checked as well and updated if it is not correct any more. Keep in mind that due to the lack of true replications, any such correlations have a low statistical power and are highly susceptible to Type 1 errors; even just 3 or 4 varieties with very high or very low values can affect the correlation substantially.
Reference 40 is not about mulberry or DNJ.
Lines 440-450: Remove all the antioxidant stuff from this paragraph; it is not necessary, and detracts from the novelty and scientific value of the paper (as explained regarding the introduction) rather than adding to it.
Lines 451-461: Just delete, for the same reasons as above.
Somewhere in the discussion it would be good (but not an absolute requirement, since it is evident from the revised title) to explain the study's limitations and opportunities: That the present study was conducted for only one year and without full replications. And that the substantial differences it demonstrated, warrant additional trials in future years, to determine to what extent the effects are constant across different years or locations, as well as relevant human or animal trials to determine the effects on health.
Conclusion:
For the reasons explained regarding the introduction, in line 464 replace 'individuals who need' with 'researchers investigating treatments'; in line 468 replace 'individuals seeking' with 'researchers seeking ways'; in lines 468-469, replace 'indi-viduals seeking' with 'researchers investigating'; in line 471, insert 'may' between 'but' and 'also'.
Consider to include an invitation to researchers who want to aquire some well-characterised plant material from your research station, which the researchers can use to carry out research on health effects. This is optional, not a requirement for publication, however it might increase the number of citations.
Please note that an article which inspires a lot of researchers to investigate a product, which subsequently becomes accepted by the scientific community, will get lots of citations and may become quite famous.
In contrast, a paper that makes unsubstantiated health claims (like the present version of the manuscript) may cause legal problems for companies, which may lose a lot of money or even end up in court, if they trust the recommendations too much. And if this happens, the paper will reduce the reputation of both the authors and the journal it is published in! It may be retracted from the journal or even cause disciplinary problems for the authors. The higher the reputation of the journal, the greater the potential risk for such damage; this is the reason why such papers are not normally published in Q1 journals.
Author Response
The manuscript is very much improved, and it is appreciated that the authors express understanding of some of the limitations of the study's design. However, additional changes are still necessary before it becomes suitable for publication.
It is essential that the limitations of the present study (of being unreplicated and not including human trials other than sensory evaluation) must be more clearly shown in the manuscript, not only in the response to reviewers.
The title must be changed accordingly; for example, an acceptable title would be: 'Preliminary Comparisons of Tender Shoots and Young Leaves of 12 Mulberry Varieties as Vegetables and Constituents Relevant for their Potential use as Functional Food for Blood Sugar Control'. This title describes the data that are actually in the manuscript, not only speculations about what this knowledge could be used for, while still indicating the key aspect. The word 'Preliminary' may be replaced with another word with similar meaning, like 'Pilot' or 'Exploratory'.
Answer: The title has been changed follow the opinion from reviewer.
Abstract
In line 20 delete this text: 'found to be the primary contributor to mulberry leaf antioxidant activity and', because the design does not allow determination of which compound (s) contributed to the antioxidant activity, only which one was most strongly associated.
Answer: The text: 'found to be the primary contributor to mulberry leaf antioxidant activity and' was deleted follow the opinion from reviewer.
Between lines 22 and 23 (after 'for' and before 'blood'), insert 'development as a functional food for', because much more research is needed before a functional food could be registered. Also, it is not clear why a higher sugar content would be superior for blood sugar control, since products for blood sugar control are usually chosen for the lowest sugar content. It might still be true, but this requires a human intervention trial to find out.
Answer: Modified the sentence of lines 22 and 23 follow the opinion from reviewer.
In line 25, delete 'individuals making informed' and replace it with 'future research into', because none of these varieties have been tested for blood sugar control in an appropriately designed and replicated trial.
Answer: The text of 'individuals making informed' in line 25 was deleted, and replace it with 'future research into' follow the opinion from reviewer.
Introduction
The question of antioxidant effect is not about whether researchers and the public believe in the health benefit, it is about what is good or bad science. The paper doi:10.1001/jama.298.21.2517 shows how researchers with limited expertise on human health research continued to cite their favourite 'discovery' papers for many years after the 'discovery' was disproved! However, this was only in low-impact journals, since the high-impact journals rejected those papers. 'Plants' is a Q1 journal, so should not accept manuscripts explicitly basing their research on disproved evidence. This paper doi:10.1001/jama.2013.277028 shows that on average, antioxidant supplementation increases mortality rather than reducing it. Do you, as an author of this manuscript, really want to take responsibility for promoting a myth, which is continuosly exploited by unscrupulous supplement manufacturers, who are killing people with their products? Or do you want to contribute to research into healthy vegetables that actually improve human health?
Most of the data on antioxidant activity in the manuscript relates to phenolic antioxidants, which probably do not kill the consumers, but where any measurable reproducible benefits have still not been documented, despite thousands of research studies attempting to do so. Regarding product quality, a high level of phenolic antioxidants is indeed important for the taste and appearance of a fresh vegetable; however, the effect can be either beneficial or harmful depending on the product and how it is cooked/prepared before consumption. The key point in relation to the present manuscript is that the important and unique aspects of this work are the measurements of 1-Deoxynojirimycin together with a range of more conventional vegetable quality characteristics. There is no need to include unsubstantiated claims about antioxidants and human health. The paper will be equally informative to the readers if the introduction instead states something that is neutral and true, like that 'antioxidant activity is a frequently reported quality measure for vegetables', rather than presenting this as a general benefit.
In contrast, the introduction should provide more details on the studies reported in the recent papers on mulberry shoots and blood sugar control that are referenced in the introduction, instead of the antioxidant information. This work is recent and interesting, it has not been disproved and is directly relevant for the manuscript. There is no maximum limit to the length of the manuscript, so it would be fine to use a paragraph or two on this to educate the readers. However, it is also important to clearly specify (see https://www.gov.uk/government/publications/uknhcc-scientific-opinion-white-mulberry-leaf-extract-and-blood-glucose-levels/scientific-opinion-for-the-substantiation-of-a-health-claim-on-a-single-component-of-morus-alba-white-mulberry-leaf-extract-and-assisting-healthy-bl) that the evidence for the benefit of mulberry leaves is still not sufficiently firm to allow its use as a functional food in Europe. The regulatory rejection is not a disproval, only a request for more and better research data; but the results in the present manuscript can only be implemented globally after a sufficient volume of such additional research has been completed and demonstrated a consistent effect.
Remove the mention of carotenes in the introduction, since this group of compounds was not measured, and also did not contribute to the antioxidant assays, since they are not soluble in 80% methanol or buffer.
Answer: The mention of carotenes in the introduction was removed follow reviewer’s opinion.
The reference to gluconsinolates is relevant, however, while tomatoes definitely have health benefits for humans, there is no evidence that this is caused by lycopene, and this is not even the subject of the reference. In contrast there is plenty of evidence that it is caused by 'something else' in the tomatoes, even though we still don't know what the active component is! See this review: doi: 10.3389/fnut.2022.849841 (which also gives useful information about what is required before a food can be approved as 'functional' in Europe) and the clear outcome of this small animal trial: doi: 10.1007/s00394-007-0687-2. For examples of plant constituents with well-documented human health effects, better choices would be other vegetables/herbs such as garlic, ginseng, onion etc.
References 5 and 6 are not about anti-inflammatory effects of flavonoids, please replace with relevant references or remove/revise this sentence completely. Those references do appear to be relevant for the description of research on DNJ so may be used in that context instead.
Answer: Replaced new relevant references about anti-inflammatory effects of flavonoids for references 5 and 6.
Lines 83-84: EITHER remove the text 'not only offer individuals with high blood sugar levels the opportunity to make informed choices regarding their vegetable selection, but also' OR replace it with some mention of the potential for future approval as functional foods, WITHOUT giving the impression that your best performing variety is ready for this type of use. The 'informed choices' by 'individuals' is wrong, BOTH because of the lack of replications in the present trial, AND because of the still not sufficient human trial evidence. Both of these deficiencies can be alleviated by future research (and hopefully will do this soon), so this change does not detract from the manuscript, as long as the manuscript avoids making unsubstantiated promises.
Answer: The text 'not only offer individuals with high blood sugar levels the opportunity to make informed choices regarding their vegetable selection, but also' was removed.
Methods
Overall organisation: The methods corresponding to each of the results should be shown in the same sequence; at present, the sensory testing comes before phenolics in the Methods, but after in the Results. Please re-arrange either the Methods section or the Results section, so the sequecne of the subsections become consistent with each other. This also applies to the Discussion section, although it is not as important there.
Answer: The methods section and the relevant references was rearranged in modified manuscript.
Regarding sample preparation: Some of the samples were freeze dried (on pages 3, 4 and 5); include a small section with the details of the freeze drying method, in particular the equipment used and the maximum temperature reached during the process.
Answer: The freeze-dried method for sample treatments was added in the section 2.1.
The sensory testing is now much more clearly described, in lines 115-118 specifying the analytical scales actually used, this is fine. But why does it still mention a hedonic scale in lines 109-110? Maybe a 'Likert scale' (with numbered points and a text equivalent for each point) was mislabelled as a 'hedonic scale' (with like/dislike text)? This is a common error, because Likert scales are often used with hedonic text. An analytical scale is a scale where the panellists are asked to assess the properties of a product, while a hedonic scale assessess the panelist's liking of the product. Check the literature for references on 'Likert scales' and 'Hedonic scales' and correct the text if relevant. If a hedonic scale was also used, then include the results in Table 2 as 'Preference'.
In lines 118-119 it says that the average scores were calculated, which is fine; however in Table 2, the values for 'Overall score' are not the average of the other scores. Find out where the error is and correct it.
Answer: The word “Overall” was detected, and the score was calculated by SFVM, namely the “average score”.
In section 2.3 (the colorimetric analyses), there is only a reference for one of the 4 methods shown. The methods are defined by their names, which would be OK if they were done using so well-established and previously validated methods that no reference was necessary. However, the anthocyanin method described is very different from the established standard method with the same name (AOAC 2005.02: Total Monomeric Anthocyanin Pigment Content of Fruit Juices, Beverages, Natural Colorants, and Wines- pH Differential Method). EITHER provide the full reference to the article showing the validation and details of the method, OR consider to drop any mention of anthocyanins, if the reported values are not comparable to values measured using validated methods.
In line 163, the word 'properly' is not clear; check if there is a typo or other error? The content of the sentence is clear and appropriate, this just looks like a language error.
Answer: This error was checked and modified in line 163.
The DPPH assay also seems to lack a reference.
Answer: Added the reference (new reference 20) for DPPH assay in in modified manuscript.
The box plots mentioned in line 228 are missing from the results. The word 'Evaluations' at the end of this line makes no sense; should it be 'Correlations'?
Answer: The expression of “box plots” was an express error, and has been detected from manuscript. “Evaluations” was the sensory evaluations, and this sentence was improved.
Results
Even though it is clearly stated in line 224 that all values are shown as Mean±SD, then this must be stated again in every table or figure legend.
Answer: Mean±SD have been stated again in every table or figure legend.
If a table (like Table 2) or figure (like Figure 3) uses letters to present significant differences, then this must also be explained (in a few words) in the table heading or a footnote or figure legend. This explanation MUST specify if the significance was calculated separately for shoots and for leaves (which is correct), or with all these data mixed together (which would be incorrect) AND which method was used to control the type 1 error (for example the Tukey test).
Answer: The test method of significant differences was explained in the footnote, figure legend, and method section.
As previously mentioned, it is best (albeit not essential) to report the concentrations of the constituents per 100g fresh weight. It is acceptable to report some of them (like the antioxidant measures) as µmol/g DW, since this is common in other papers.
Answer: Thanks to reviewer’s good suggestion for us, we will express the data more reasonably in future research about mulberry leaves vegetables.
The anthocyanin values in Table 1 are surprisingly high, and the differences do not match the colours shown on Figure 1. For example, the paper https://doi.org/10.3389/fpls.2023.1155722 showed that lettuce cultivars with red-coloured leaves contained between 0.2 and 1 mg/g DW anthocyanin. The leaves on Figure 1 are only slightly mauve in colour, much lighter than red lettuce leaves, so would be expected to contain less that 0.2 mg/g DW, according to this article. As mentioned before, the corresponding method description is also rather strange, so it is possible that the values are simply not correct. Since it is clear from the photos that the anthocyanin content is very low, the manuscript will not lose any important information if all mentions of anthocyanins are removed; this is better than including values that are not measured with a correct method.
Answer: This result may be influenced by the test method, and we will use more precise methods in future research.
For vitamin C, which was measured on fresh plant material, it is absurd to report the value as mg/100g DW; hopefully this was just a typo, and it actually means mg/100g FW! if so please correct the typo.
Answer: The typo of “mg/100g DW” was corrected as “mg/100g FW”.
If it is not a typo, and the authors explicitly want to use only DW values for the reporting, then do it as mg/g DW, so at least it is comparable with the other values in the present study.
Answer: The typo of “mg/100g DW” was corrected as “mg/100g FW”.
Note that if the mg/100g DW is NOT a typo, then the values for vitamin C are low (compared with other vegetables), and it is then NOT a rich source. In this case, correct the statement in line 295.
Answer: The typo of “mg/100g DW” was corrected as “mg/100g FW”, therefore, the statement in line 295 was not corrected.
In any case, no matter which units are used, it is absolutely essential to report the Dry Matter percentage (gDM/100g FW) for each of the 24 samples, in one of the tables; with this information, the reader can compare the values in the present study with values from the literature, irrespective of which unit it is reported in.
Answer: Thanks to reviewer’s good suggestion for us, we will express the data more reasonably in future research about mulberry leaves vegetables.
The term 'score' is used both for antioxidant activity and for overall sensory evaluation. Make sure it is clear in every case which type of score it is about; in particular in Figure 5 and 6 and their figure legends.
Answer:
Line 312: the text is clear but not correct English, please improve.
Answer: The text in line 312 was improved in modified manuscript.
In Table 2, the SD, Mean, SD between varieties and CV between varieties are missing for the Overall score. As mentioned above, the values of the Overall score also do not match the mean of the other scores, as they should according to the method description. Please find out what the real values are, fix either the values or the method description so they match, and show all of the values; or, if they are an error that cannot be fixed, remove them so they don't confuse the reader.
Answer: The Mean, SD between varieties and CV between varieties for Overall score are added in the Table 2.
The correlations between VC and DNJ are shown on both Figures 5 and 6, which is fine; but why is the value for the tender shoots different in the two figures (0.16 in Figure 5 and -0.44 in Figure 6)? For the leaves, the value is the same (-0.20) in both figures.
Answer: An error annotation in Figure 6, and we have modified the Figure 5 and Figure 6.
The text in lines 331-335 does not match the values shown on Figure 5! In particular in the tender shoots, the DNJ values on the figure are not correlated with anything, while the text says that the correlation is very high (0.63). Which is correct?
Answer: A correct Figure 5 was added in the modified manuscript.
Please check that there are no errors in all the data in all the tables and figures! If the values are correct, then explain why the two figures gave different results, for example if there are different numbers of samples in separate calculations, and if so, what they are.
Answer: All the data in all the tables and figures were checked in the modified manuscript.
Discussion
Lines 412-413: Delete this sentence, for the reasons explained in the introduction. The authors of those references know much less about this topic than the experts at EFSA.
Answer: The sentence of lines 412-413 was deleted follow the opinion from reviewer.
Line 426-27: Remove the word 'alarmingly', it is unnecessary, unsubstantiated and confusing when not explained in context.
Answer: The word 'alarmingly' in line 426-27 was removed.
Line 435: Remove the word 'vast', it is unnecessary, unsubstantiated and confusing when not explained in context. Also correct the grammar (singular/plural) of this sentence.
Answer: The word 'vast' in line 435 was removed, and the sentence was modified.
Line 437: the text is clear but not correct English, please improve.
Answer: The sentence of line 437 was modified.
Lines 443-444: Figure 5 shows little or no correlations between DNJ and antioxidant activities, despite stating the opposite in the text. Once the errors in the data have been found and corrected, so the results have changed, then the discussion must be checked as well and updated if it is not correct any more. Keep in mind that due to the lack of true replications, any such correlations have a low statistical power and are highly susceptible to Type 1 errors; even just 3 or 4 varieties with very high or very low values can affect the correlation substantially.
Reference 40 is not about mulberry or DNJ.
Answer: A new Reference 40 about mulberry and DNJ have been changed in manuscript.
Lines 440-450: Remove all the antioxidant stuff from this paragraph; it is not necessary, and detracts from the novelty and scientific value of the paper (as explained regarding the introduction) rather than adding to it.
Answer:
Lines 451-461: Just delete, for the same reasons as above.
Somewhere in the discussion it would be good (but not an absolute requirement, since it is evident from the revised title) to explain the study's limitations and opportunities: That the present study was conducted for only one year and without full replications. And that the substantial differences it demonstrated, warrant additional trials in future years, to determine to what extent the effects are constant across different years or locations, as well as relevant human or animal trials to determine the effects on health.
Answer: Lines 451-461 was deleted follow the opinion from reviewer.
Conclusion:
For the reasons explained regarding the introduction, in line 464 replace 'individuals who need' with 'researchers investigating treatments'; in line 468 replace 'individuals seeking' with 'researchers seeking ways'; in lines 468-469, replace 'indi-viduals seeking' with 'researchers investigating'; in line 471, insert 'may' between 'but' and 'also'.
Answer: Modified the sentences follow the opinion from reviewer.
Consider to include an invitation to researchers who want to aquire some well-characterised plant material from your research station, which the researchers can use to carry out research on health effects. This is optional, not a requirement for publication, however it might increase the number of citations.
Please note that an article which inspires a lot of researchers to investigate a product, which subsequently becomes accepted by the scientific community, will get lots of citations and may become quite famous.
In contrast, a paper that makes unsubstantiated health claims (like the present version of the manuscript) may cause legal problems for companies, which may lose a lot of money or even end up in court, if they trust the recommendations too much. And if this happens, the paper will reduce the reputation of both the authors and the journal it is published in! It may be retracted from the journal or even cause disciplinary problems for the authors. The higher the reputation of the journal, the greater the potential risk for such damage; this is the reason why such papers are not normally published in Q1 journals.
Reviewer 2 Report
The authors have addressed almost all the raised comments. Therefore, I recommend the paper for publication.
Minor editing of English language required.
Author Response
English language editing for our manuscrpt will be arranged by MDPI .
Reviewer 3 Report
After review the revised version of the manuscript I can see that the Authors have done a lot of work to improve the quality of the article. Undoubtedly, this contributed to achieving the assumed goal - increasing the readability of the article. The suggested corrections were made. According to me, this work is an interesting study and I recommend publishing it in the "Plants" journal.
Author Response
English language editing for our manuscrpt will be arranged by MDPI